# Intrinsic neuronal properties represent song and error in zebra finch vocal learning

Arij Daou [1,2] & Daniel Margoliash [1✉]

Neurons regulate their intrinsic physiological properties, which could influence network properties and contribute to behavioral plasticity. Recording from adult zebra finch brain slices we show that within each bird basal ganglia Area X–projecting (HVC$_X$) neurons share similar spike waveform morphology and timing of spike trains, with modeling indicating similar magnitudes of five principal ion currents. These properties vary among birds in lawful relation to acoustic similarity of the birds' songs, with adult sibling pairs (same songs) sharing similar waveforms and spiking characteristics. The properties are maintained dynamically: HVC$_X$ within juveniles learning to sing show variable properties, whereas the uniformity rapidly degrades within hours in adults singing while exposed to abnormal (delayed) auditory feedback. Thus, within individual birds the population of current magnitudes covary over the arc of development, while rapidly responding to changes in feedback (in adults). This identifies network interactions with intrinsic properties that affect information storage and processing of learned vocalizations.

[1] Department of Organismal Biology & Anatomy, University of Chicago, 1027 E. 57th St., Chicago, IL 60637, USA. [2] Present address: Biomedical Engineering Program, American University of Beirut, P.O. Box 11-0236, Riad El Solh, Beirut 1107 2020, Lebanon ✉email: dan@bigbird.uchicago.edu

Whereas memories are widely thought to be implemented by plasticity in synaptic strength mediated via activity-dependent changes[1], non-synaptic forms of plasticity have also been implicated[2–4]. The type and magnitude of ion currents that a neuron expresses contribute to the number, timing, and patterns of action potentials generated in response to a given input, hence the neuron's contribution to network dynamics[5–7]. Plasticity of these intrinsic properties (IP) includes homeostatic and other forms of regulation[5,8,9], suggesting the potential for widespread contribution to brain plasticity. Neuronal intrinsic properties have been manipulated in vitro and in vivo, by numerous experimental approaches but also including behavioral conditioning[10,11], may involve one or more ion currents[8,12], and can be limited to segments of dendrites[13,14], or the somatic[2] or axon hillock[15,16] compartments. Changes in the latter compartments should modulate currents that arise from all processes of spatiotemporal summation in the neuron's dendrites. Yet, relatively few studies have explored how such potentially widespread and powerful plastic mechanisms are expressed in complex learned behaviors beyond operant conditioning, developmentally[17,18] and in populations of adult neurons.

Birdsong learning is a well-established model for complex vocal learning. Song learning is regulated by auditory feedback, essential for song development in juveniles[19] and song maintenance in adults[20]. Manipulating either spectral or temporal features of auditory feedback induces changes in singing behavior, and birds can make adaptive responses depending on the parameters and specific song syllable that receives modified feedback. Understanding the mechanisms of such rapid and precise feedback regulation is important for informing computational theories of internal models of behavior.

Adult zebra finches present an interesting model, in that they sing exceedingly precise and invariant songs especially when directing them toward females and are thought to be relatively insensitive on short time scales to feedback perturbations[21], unlike other species such as Bengalese finches[22]. Yet, zebra finches require auditory feedback to maintain their songs[20]. What mechanistic differences give rise to these different patterns of feedback regulation in different species?

The forebrain song system includes a motor pathway that ultimately engages syringeal and respiratory muscles to produce song, and a basal ganglia pathway that is engaged in feedback-mediated song learning and maintenance. Within the forebrain nucleus HVC (the proper name), the basal ganglia Area X-projecting ($HVC_X$) neurons projecting to the basal ganglia component of the song system (Fig. 1a) show precisely timed activity during singing[23] perhaps carrying information about singing while not being required for singing[24]. In quiescent birds highly song-selective auditory responses in HVC neurons have been observed in several species of songbirds[25,26] but these are greatly suppressed in awake zebra finches[27–29]. Furthermore, in response to auditory feedback perturbation during singing HVC spiking activity does not change in zebra finches[30] including at $HVC_X$ synapses[31,32], whereas similar manipulations reliably change HVC activity in Bengalese finches[33]. Perhaps this pattern of feedback insensitivity in zebra finches facilitates production of ballistic (feedback invariant) singing, but if so how do the birds evaluate feedback? Here, we report a surprising mechanism that stores information about song structure rapidly. By analyzing the raw data traces in response to somatic current injections, we show a consistent pattern of homogeneity of $HVC_X$ intrinsic properties within individual adult birds and variation among birds related to the birds' songs. The homogeneity is maintained by plastic mechanisms and is sensitive to auditory perturbations.

## Results

**Individual birds have uniform $HVC_X$ intrinsic properties.** We made intracellular whole-cell patch visually guided current clamp recordings in a brain slice preparation (see Methods: Slice preparation & Whole-cell recordings) from a total of 370 $HVC_X$ neurons distributed across multiple experimental designs in 76 animals. We unambiguously identified $HVC_X$ by their distinct physiological properties, as described previously[34–38], and confirmed these offline. Physiological characterization was confirmed in 70% (259/370) of neurons filled with Biocytin (see Fig. 1b & Methods: Histology), of which 96% (248/259) had axons we could visualize, all of which were clearly projecting toward Area X (Fig. 1a, b). Physiological characterization of $HVC_X$ neurons was further confirmed, without exception, in 14 adults and 1 juvenile, in recordings from 114 cells that were retrogradely labeled by tracer injections into Area X (see Methods: Retrograde tracers). We characterized every cell for which we achieved stable recordings. Once the structure of our results began to emerge, we focused on recording as many $HVC_X$ neurons per animal as possible, began to record the song of each bird prior to making the slice recordings, and systematically brought all cells to −70 mV before passing stimulating currents, to minimize variability arising from voltage-dependent changes in membrane conductances (resting membrane potential at break-in was −70.49 ± 2.3 mV (mean ± SD here and throughout the paper), current to hold cells at −70 mV: 9.73 ± 16.35 pA, min −30 pA; max 35 pA; $N = 370$ $HVC_X$). All cells received canonical depolarizing (100 pA, 200 ms) and hyperpolarizing (−140 pA, 200 ms) current pulses, as well as many other stimuli.

$HVC_X$ emit brief intense bursts of spikes during singing[30], and emit adapting spike trains in slice when stimulated by somatic current injection[34,36]. In response to injections of the canonical depolarizing currents, we observed that the spike patterns for the $HVC_X$ of a given bird all tended to have similar onsets, numbers of spikes, and timing of spikes, whereas these characteristics varied from bird to bird (Fig. 1c). A plot of the first interspike interval (ISI) versus spike rate showed a pattern of within-bird homogeneity and across-birds heterogeneity for the population of $HVC_X$ ($N = 253$ neurons arising from all 51 adult birds each with two or more $HVC_X$, but excluding the second of a sibling pair or post-pharmacological and other experimental treatments (see below), 4.96 ± 2.31 neurons/bird), and individual birds were highly significantly different from the population means (Fig. 1d). The hypothesis that $HVC_X$ in all birds had the same means of first ISI, number of spikes, and spike rate was rejected (MANOVA: see Methods, MANOVA analysis, Wilks' lambda = 0.0001, $F = 86$, d.f. = 51, $P < 1.5 \times 10^{-5}$). Under a more stringent set of null hypotheses in which each pair of birds shared the same mean, 69% (628) of the pairwise MANOVA comparisons were rejected at the 5% level (after using the conservative Bonferroni correction). In a principal components analysis of spike frequency and the first three ISIs, $PC_1$ accounted for 90% of the variance across birds (Supplementary Fig. 1). Regulation of the number and rate of spikes $HVC_X$ neurons emit will affect their contribution to the basal ganglia pathway. Thus, in these slice recordings, the activity that $HVC_X$ neurons transmit to their network was relatively uniform within each bird and varied far more from bird to bird. Such uniformity within individual animals was not known, and was unanticipated given the variation within individual birds in the spiking patterns of different $HVC_X$ observed during singing[30].

The spike waveforms of the $HVC_X$ also tended to conform to a common shape for a given bird, while spike shape varied from bird to bird (see Supplementary Fig. 1 for definitions of all spike waveform features we characterized). We characterized this by examining the shapes of the first spikes in a burst, one spike from

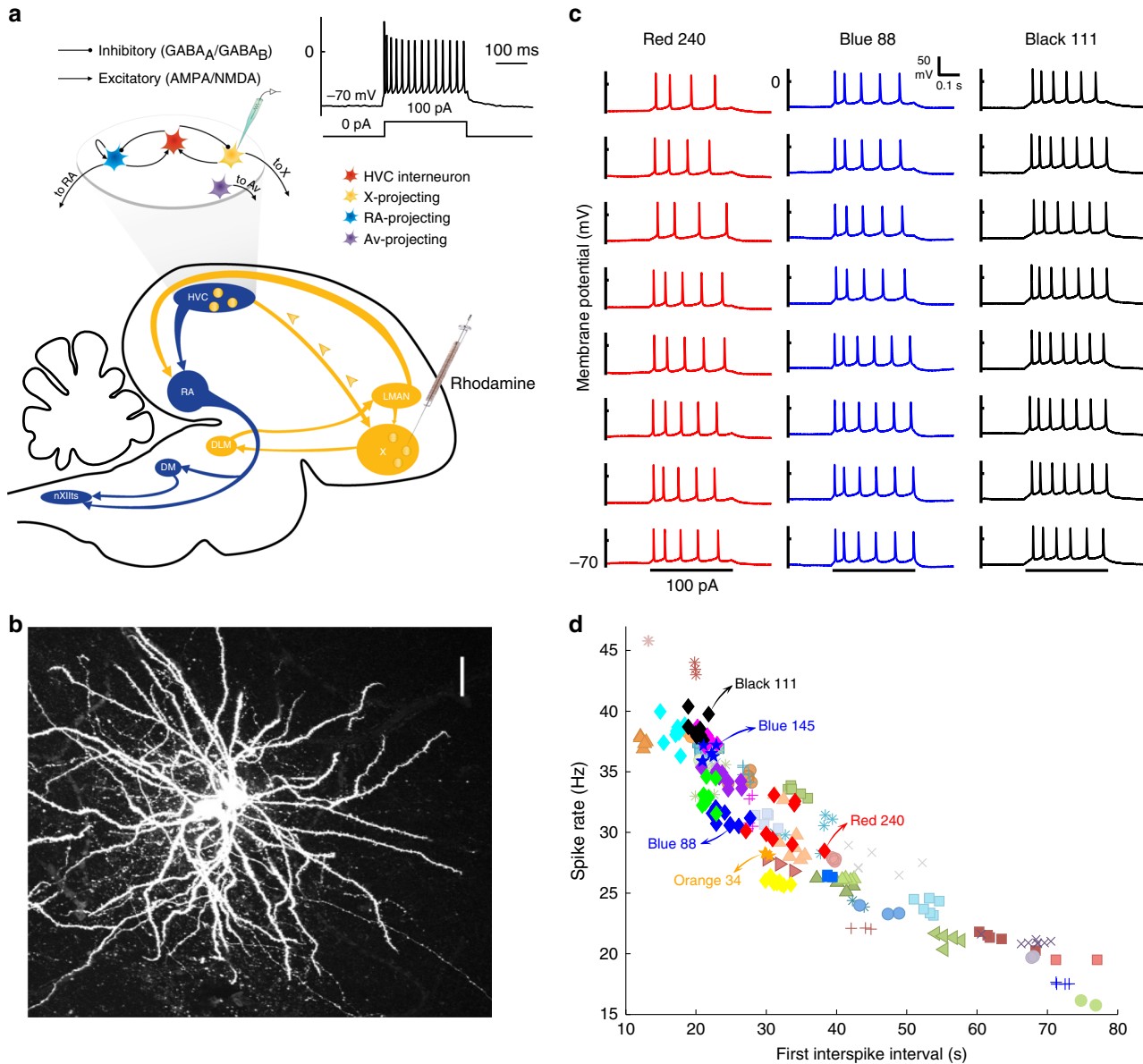

**Fig. 1 Individual birds have uniform and distinct HVC$_X$ intrinsic properties: spike trains. a** Song system, experimental setup and whole-cell recordings. Schematic of the song system (sagittal view) with schematic of HVC cell types in a slice, above. HVC includes interneurons (HVC$_I$), HVC$_{RA}$ neurons projecting to the robust nucleus of arcopallium (RA) of the motor pathway, and HVC$_X$ neurons projecting to Area X. (Few HVC neurons project both to RA and Area X[63]; none of our fills showed such bifurcating axons, nor did we fill any neurons projecting to nucleus Avalanche, Av[64].) HVC$_X$ neurons were pre-labeling with tracer injections into Area X. Inset shows exemplar HVC$_X$ voltage response to current injection. Note the spike train rides on a plateau voltage and the spike rate accommodates over time. These are two of the many features of HVC$_X$ response that were consistent within each bird but that varied from bird to bird. **b** Example of labeling HVC$_X$ neurons. The axon of each cell can be seen in the lower right, which is projecting toward Area X. Scale bar: 30 μm. **c** Eight distinct neurons from each of three birds (Red 240, Blue 88, Black 111). Spiking patterns in response to 100 pA, 200 ms depolarizing current pulses from all eight HVC$_X$ from each of the birds, showing within-bird homogeneity and across-birds heterogeneity. Traces are sorted top to bottom in increasing order of number of spikes and then duration of first ISI; birds are sorted left to right in order of increasing excitability. **d** Scatter plot of first ISI versus spike rate showing same homogeneity/heterogeneity effect in population data ($N = 253$ neurons). Each of $N = 51$ unique color/symbols represents the data from one bird. Diamond symbols represent birds with 8 or more recorded neurons. Arrows with text point identify the birds shown in Figs. 1–3 that are referred to in the text, including the pairs of birds recorded on the same day (Black 111/Red 240 and Blue 145/Orange 34). Each color/symbol refers to the same bird in Fig. 1d, Fig. 2c, Fig. 3c, and Supplementary Figs. 3, 7.

each cell, in response to the canonical depolarizing stimulation. In three birds with the greatest numbers of HVC$_X$ recorded ($N = 10$, 9, 8), the cells from a given bird had strikingly similar waveforms, whereas each bird was associated with a different spike waveform shape (Fig. 2a). For the eight birds with the largest numbers of HVC$_X$ recorded ($n \geq 8$), comparing the means of the first spike waveforms helps to illustrate the variation across birds including

variation in the mean spike amplitudes and spike thresholds (Fig. 2b). Spike amplitude and spike threshold segregated neurons from different birds into visually highly distinct clusters (Fig. 2c), and the clustering was highly statistically significant (see Methods: Additional statistical analysis, Fig. 2c). Spike threshold is thought to reflect excitability in the soma and near the spike initiating zone[39]. Thus, in each bird the population of HVC$_X$

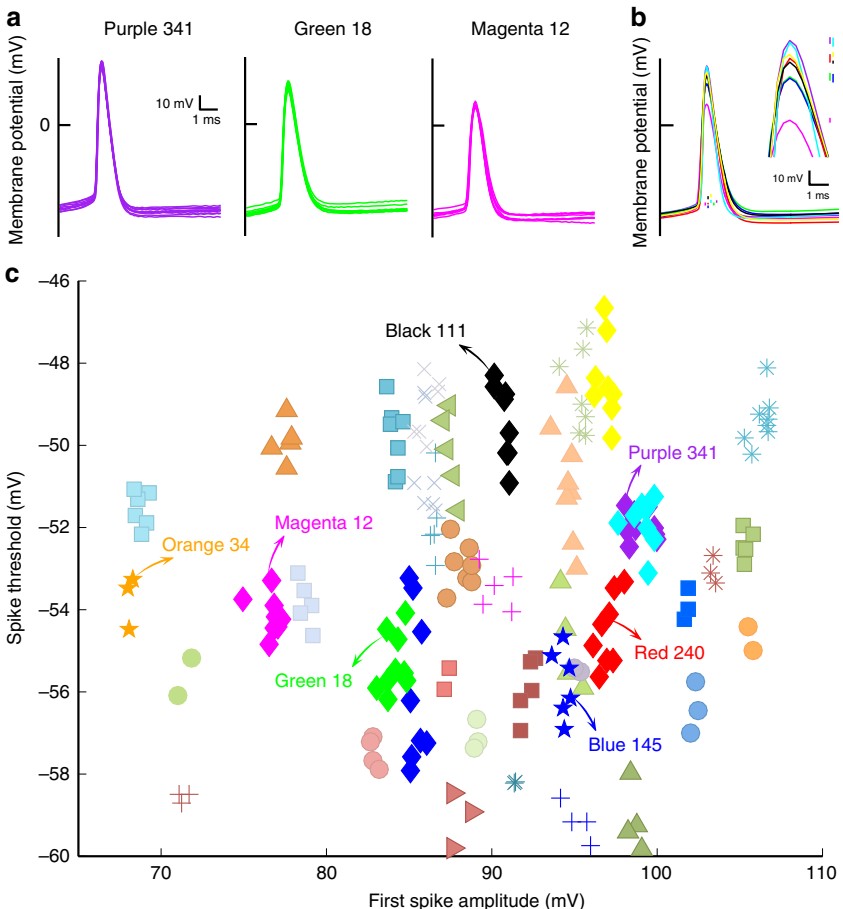

**Fig. 2 Individual birds have uniform and distinct HVC_X intrinsic properties: spike waveforms.** The same color is used to represent birds that appear in more than one panel. **a** The first spike waveforms have similar shape within each bird (Purple 341, 10 HVC_X; Green 18, 9 HVC_X; Magenta 12, 8 HVC_X), and spike shapes differ across the birds. **b** The averaged first spike waveforms for each of the eight birds with the greatest numbers of HVC_X recorded (three birds from Fig. 1c, three birds from Fig. 2a, two other birds with 8 HVC_X). Each bird is represented by a different color. Vertical bars are centered on the spike threshold values, with the height of the bars representing the spike threshold variability (±1 SD). The bars are slightly offset from each other to improve visibility. Inset shows detail of average spike waveforms near their peaks. The vertical bars are centered on the peak of the average spike amplitudes, with the height of the bar representing the amplitude variability (±1 SD). **c** Neurons tend to cluster in distinct bird-specific regions in the space of spike amplitude and spike threshold.

exhibited homogeneity both in spike timing and spike morphology, and these features varied from bird to bird. Numerous other features of IP also clustered neurons from different birds (Supplementary Fig. 3). Features such as spike amplitude, plateau amplitude, spike threshold, and interspike interval should vary with input resistance. We found such relations in our data, for example, input resistance ($181 \pm 35\,M\Omega$, $N = 253$ neurons) was significantly related to first spike amplitude whether we considered all neurons ($R^2 = 0.29$, $P = 4 \times 10^{-6}$, $t$-test), or the average across cells of the input resistance and first spike amplitudes for each of the 51 birds ($R^2 = 0.35$, $P = 0.012$, $t$-test).

Control experiments and analyses rejected several hypotheses that the variation among birds arose from uncontrolled variation in the experimental conditions. On each of 2 days we recorded from two pairs of birds (Red 240/Black 111; Blue 145/Orange 34) using the same ACSF and intracellular solutions for both birds of each pair. The HVC_X spike train values (ISI versus spike rate) for the individual neurons showed clustering within each of the four birds but varied greatly across the birds (Fig. 1d), and the spike waveforms of neurons from each bird were similar but these too varied greatly between the birds of a given pair (Fig. 2b; Supplemental Fig. 3). This demonstrates that factors other than osmolality, internal (pipette) solution or relative ion

concentrations of the ACSF prepared on the day of the experiment gave rise to variation between birds in the HVC_X IP. Also, there was no time-dependent variation in the IP values comparing the spike amplitudes of different cells recorded over the duration of an experiment or comparing the average spike amplitudes for each bird over the longer intervals (days) between experiments (see Methods: Rundown). There was also no relation between series resistance and first spike amplitude ($N = 253$ neurons, 51 birds, $R^2 = 0.029$, $P = 0.65$, $t$-test). Thus, we found no evidence that non-specific effects explained the results, motivating a search for biological mechanistic and behavioral explanations for the variation of HVC_X IP.

**Modeling the ionic basis of uniform intrinsic properties.** The relation between neuronal dynamics and spike shape should be influenced by the magnitude of ion currents of the given cell. We explored the relation between HVC_X dynamics and spike shape by modeling the magnitude of ion currents in a large subset of the neurons (290 HVC_X in 63 birds, all cells modeled for any given bird), including experimental birds (see below) and a large subset of the non-manipulated control birds (151 neurons, 38 birds, birds chosen at random). Cells were modeled with a charge

balance equation of the Hodgkin–Huxley type (see Methods: HH model fitting), using an HH model incorporating ion currents that have recently been pharmacologically demonstrated for $HVC_X$[34]. We used a single-compartment model, thus not differentiating potential contributions of dendritic or somatic ion currents to the somatic measurements that we made and modeled. Most parameters of the HH model were held constant at biologically realistic values, while we varied the magnitudes of the maximal conductances of five principal ion currents including fast sodium channels ($I_{Na}$), potassium channels ($I_K$), calcium-dependent potassium channels ($I_{SK}$), hyperpolarization-activated cyclic nucleotide-gated channels ($I_h$), and T-type calcium

channels ($I_{Ca-T}$). Modeling consisted of manually adjusting the values of the five current magnitudes to achieve a good fit between the output of the HH equation (model neuron) and the actual neuron's behavior, only considering responses to the canonical depolarizing and hyperpolarizing current injections[34]. (The effects of the ion currents on $HVC_X$ spike waveform features has been extensively described elsewhere[34].) A good fit was judged based on heuristics examining the spike shape and spike train, and in many cases, excellent fits were obtained (Fig. 3a, left panel).

Model fits were tested by assessing model predictions in response to current injections not used in the fitting process.

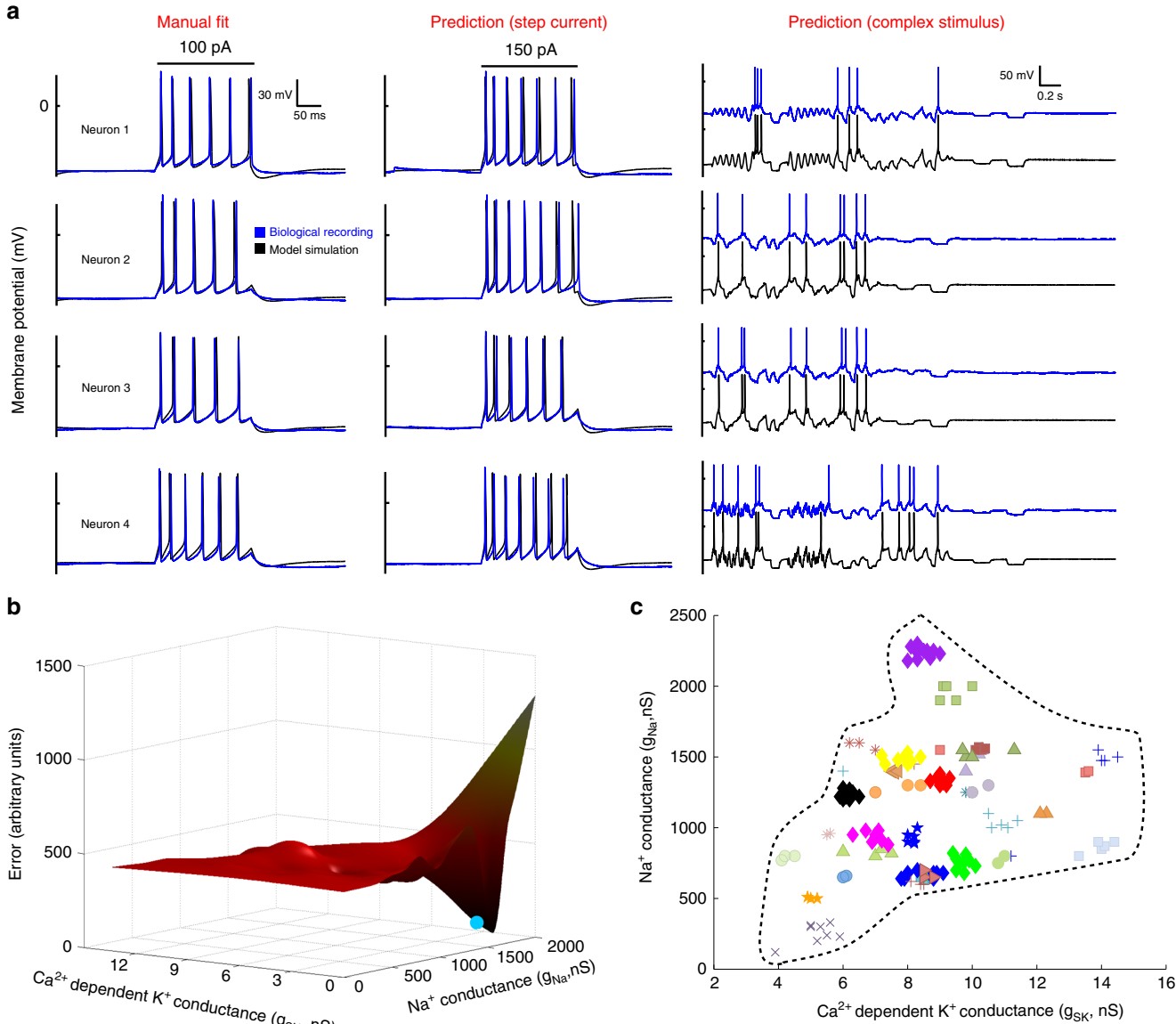

**Fig. 3 Modeling the ionic basis of uniform intrinsic properties. a** Exemplar data for four neurons where manual adjustment of the five maximal conductances achieved a good fit between the corresponding biological recording (blue traces) and the model simulation (black traces). Model fits were judged based on the response to current pulses of 100 pA (left panel). Good fits resulted in good predictions in response to current injections not used in the fitting process, for example consider the predictions to step currents of 150 pA magnitude (middle panel), and chaotic current stimuli (right panel; chaotic stimuli not shown to maintain clarity). **b** An exemplar error manifold (see text) showing two parameters ($g_{SK}$, $g_{Na}$) of the five examined in an exhaustive parameter search. The manually determined fit (blue point) resides in the same concavity and near to the global minima. **c** A scatter plot of all 151 manually modeled neurons showing the two conductances that varied the most across the population, $g_{Na}$ and $g_{SK}$. Different animals ($N = 38$), denoted by different color-symbols, occupy different regions in the 2D space. The dashed line encloses the 2D conductance space of the modeled data set, representing an estimate of the entire parameter space occupied by the population of normal adult zebra finches. The same dashed line, over the same scales, appears in Fig. 5c, 5f and 7c.

Good fits tended to result in good predictions. Good predictions were observed for step currents of other magnitudes (Fig. 3a, middle panel), as well as for chaotic currents which incorporated more complex dynamics than the step currents (Fig. 3a, right panel). Cases of good predictions demonstrate that the spike waveform shape of the neuron was linked to its bursting properties, as described by the dynamical model. To extend this analysis we adopted a quantitative measure of goodness of fit to compare model fits with model predictions, using eight parameters of response measured from the current pulse injections (see Methods: Model predictions and cross-validation). This confirmed that the models made good predictions, for example predictions for 150 pA injections for 45.7% (69/151) of cells were as good or better than that for neuron 4 of Fig. 3a. The quality of predictions at 75 and 150 pA were comparable, whereas predictions at higher currents (175 pA) were somewhat worse. We then tested whether the manual fitting procedure had found local minima or a global error minimum, which also evaluated whether the manual fitting procedure had potentially failed to identify different local minima representing different combinations of ion current magnitudes for different neurons. To this end, we conducted exhaustive parameter searches on the space of the five ion current magnitudes, assessing the output of the model against the actual neuron's output using the same measure of goodness of fit (see Methods: Exhaustive parameter searches & Supplementary Fig. 4). This demonstrated that the manual fitting produced unbiased results and tended to find global minima (Fig. 3b). Finally, we also conducted cross-validation analyses, demonstrating that the model of a given neuron was far better at predicting responses of that neuron than were models of neurons from other birds (see Methods: Model predictions and cross-validation & Supplementary Fig. 5). Whereas other parameters of the HH equations that were held constant in these modeling studies could in principle also account for similar variation within and across birds, the excellent performance of the models focused our attention on the potential biological significance of the estimated ion current magnitudes.

Examining the distribution of predicted current magnitudes (manual fitting) for the five ion channels revealed that the neurons from each animal occupied a small volume in the five-dimensional space of ion conductance magnitudes, and that there were large differences between animals. This is anticipated from the variation of spike waveforms and spike trains described above, but here is expressed in terms of magnitudes of plausible, pharmacologically confirmed $HVC_X$ ion currents. Scatter plots show this graphically for the two currents that varied the most in adult animals ($g_{Na}$ and $g_{SK}$; Fig. 3c), and for the other currents (Supplementary Fig. 6). Scatter plots comparing data fit manually and fit by parameter searches yielded similar results (Supplementary Fig. 7). In our sample, model estimates of $g_{Na}$ varied within the species by over $19.2x$ and $g_{SK}$ by over $4.1x$, while the average variation within each animal with 5 or more neurons was $1.23x \pm 0.48$ ($g_{Na}$) and $1.14x \pm 0.12$ ($g_{SK}$). Similar results obtained for the other three currents (maximum variation across all neurons/average variation within each bird: $1.24x \pm 0.15$ ($g_K$), $1.17x \pm 0.34$ ($g_h$), $1.19x \pm 0.69$ ($g_{Ca-T}$)), demonstrating each bird occupied a small space in the five-dimension space of conductance magnitude values. To quantify the joint variation for all five ion conductances for each bird, for the 8 birds with $\geq 8$ $HVC_X$, we defined the volume of the minimal hyperellipsoid spanning the distribution of all five conductance magnitudes for all neurons in each bird (see Methods: Conductance volumes). On average, a bird's hyperellipsoid occupied a volume of only $0.00037 \pm 6.1 \times 10^{-5}$ of the "species" volume, that is, the hyperellipsoid volume occupied by all 151 neurons of 38 non-manipulated birds. (If each parameter occupies 0.1 of the total range and the space is isotropic, this would yield a

volume of $10^{-5}$.) This confirms the trend that each of the five conductances had similar magnitudes for all neurons within each bird, and that these magnitudes differed greatly between birds. Considering each of the five conductance magnitude values pooled across all birds yielded distinctly non-Gaussian distributions. In a principal components analysis of the five ion conductance magnitudes, $PC_1$ accounted for 41.4% of the variance across all modeled neurons from all birds and the other PCs were around 20%. This indicates that there was no simple linear combination of these currents that captured most of the variation among animals.

**Assessing intrinsic currents in network-isolated $HVC_X$.** The modeling framework we used did not account for any current sources that potentially might arise from network activity in the slice. To control for this, we performed experiments in four animals where neurons were characterized before and after bath application of a cocktail of drugs designed to block fast synaptic transmission (see Methods: Pharmacological manipulations). (A slice was abandoned after drug treatment since washout may be incomplete, hence we recorded only one cell per slice.) In all cases, the modeling revealed a reduction in the estimated SK current while leaving the estimates for other currents unchanged (Fig. 4a). Importantly, the spike morphology of individual neurons was not altered after pharmacological treatment (Fig. 4b), even though a small increase in the neurons' resting membrane potentials was observed. Thus, isolating cells from the network did not disrupt the pattern of within-bird similarity and between-bird variation in current magnitude estimates (Fig. 4a). We interpret these results to indicate that the pharmacological treatment silenced inhibitory interneurons ($HVC_I$), which provide substantial input onto $HVC_X$[40,41] (Fig. 1a). In the HH current balance equation reduction in SK current increases the overall excitability of the cell, mimicking reduction of inhibitory input.

**Linking intrinsic cellular properties to learning.** We then turned to the powerful machinery of birdsong learning to gain insight into what factors might give rise to such unanticipated pattern of variation in the intrinsic properties of the population of $HVC_X$ neurons. Most birds were seemingly randomly distributed across the conductance magnitude parameter space (Fig. 3c). To examine if instead the distribution was related to the songs of the birds, we considered the 18 birds for which we had modeled neurons and recorded songs (see Methods: Song recordings). We calculated the Euclidean distances between the centroid of the conductance values for one bird and the centroids for each of the other 17 birds, and compared those distances to the similarities of the songs of the corresponding two birds, using a widely adopted standard measure of zebra finch song similarity (symmetrical measure comparing canonical motifs—sequences of syllables[42]). Strikingly, there was a strong correlation between the two measures, when the data for all birds were considered together (linear regression of non-identical lower triangle matrix values, $R = -0.70$, $P < 10^{-4}$, $t$-test here and below), and also for 11/18 birds when the data for each bird was considered in isolation ($P < 10^{-3}$ or smaller) (Fig. 5a). Furthermore, one outlier bird gave rise to all the non-significant results. When we remove White 20, one of two birds with the fewest numbers ($N = 3$) of $HVC_X$ recorded, all but one of the remaining birds considered individually had $P$ values < 0.001 (16 birds), with the remaining one marginally significant ($P < 0.06$), and with the group data yielding values of $R = -0.75$ and $P < 10^{-4}$. We also examined correlations between variance normalized Euclidean distances (Mahalanobis distance, see Methods: Mahalanobis distance), demonstrating a similar result (data for all birds considered together, $R = -0.75$, $P < 10^{-4}$). We also ruled out

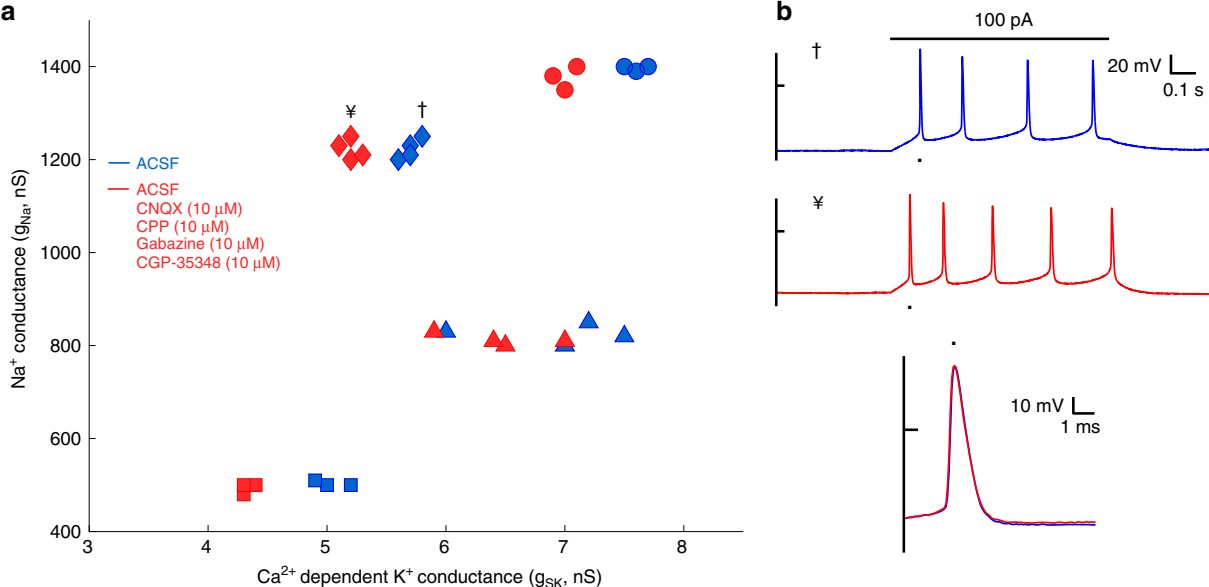

**Fig. 4 Directly measuring intrinsic currents in network-isolated HVC$_X$. a** Conductances for neurons of four birds (different symbols) before (blue) and after (red) application of fast synaptic antagonists. **b** Example neuron, before (†) and after (¥) antagonist application, shows increased excitability after synaptic blockade (top panel) but no changes in spike morphology (bottom panel).

the possibility that network effects were dominating this calculation through the contributions of the $g_{SK}$ estimates (Fig. 4) by recalculating the conductance distances after moving $g_{SK}$ (all birds including outlier, correlation of song similarity vs. modified Euclidean distance, $R = -0.69$, $P < 10^{-4}$; vs. modified Mahalanobis distance, $R = -0.54$, $P < 10^{-3}$). Thus, remarkably, the differences in the intrinsic ion currents of HVC$_X$ in different zebra finches were related to differences in the acoustic features of the birds' songs. These results motivate future work to identify which features of song best describe the variation from bird to bird in HVC$_X$ intrinsic properties, whether these are acoustic, movement-related[43], or related to other features of song performance. Such studies may yield insight into the mechanisms linking IP with sensorimotor behavior.

Song ontogeny in many songbird species continues into adulthood, well beyond sexual maturity and song crystallization. For example, in adult zebra finches sensitivity to auditory feedback (as measured by changes in song following deafening) declines with age[44]. We observed concomitant age-related changes in the variance of IP, with the five-dimensional conductance volume of space occupied by all the neurons in each bird decreasing with age (Fig. 5b). It is possible that loss of sensitivity to auditory feedback helps drive loss of variation in IP across adulthood, but future experiments are required to address whether the two are mechanistically related, and whether these relations reflect song learning or learning independent effects of aging.

**A developmental basis for homogeneous intrinsic properties.** One immediate implication of these results is that birds with similar songs should have HVC$_X$ with similar IP. We tested this by examining the IP of HVC$_X$ in four pairs of sibling adult birds. The siblings of each pair were raised in a breeding colony and had the same parents, but the four pairs arose from four different set of parents. As predicted by the relation between IP and song acoustics (Fig. 5a), neurons from each pair of sibling birds had strikingly similar spike trains (e.g., Fig. 6a) and spike waveforms (Fig. 6b), which varied among the four pairs of birds. This visual impression was supported by the distributions of measurements

taken from spike waveforms and spike trains (spike rate, first spike interval, plateau amplitude, spike threshold, first spike amplitude), with neurons from each pair of siblings tending to cluster together but in a distinct region for each sibling pair (Supplementary Fig. 8). We had access to the songs of both siblings for three of the pairs, and confirmed the corresponding prediction, that the songs of each pair of siblings shared very similar temporal and spectral features (e.g., Supplementary Fig. 9), which varied among the three pairs of songs. For each pair of siblings, the HVC$_X$ occupied very similar regions in the conductance space (Fig. 6c). This sharply contrasts with the large variation in spike waveforms, spike activity patterns, and modeled ion current magnitudes comparing pairs of randomly chosen birds (Figs. 2, 3, 4).

Sibling birds share similar genetics, similar ages, and may follow similar vocal developmental trajectories[45] to arrive at similar adult songs. To examine if vocal developmental learning contributes to regulation of neuronal ionic conductances[17,18], we made preliminary examination of the variation in HVC$_X$ populations in juvenile birds in the plastic song stage of song learning (day post hatch (dph): 55 (Juv2), 57 (JuvSib1A), 58 (JuvSib1B), 62 (Juv3)) (n ≥ 5 HVC$_X$/bird), two of which were siblings. Unlike the similarity of HVC$_X$ properties observed in each non-manipulated adult bird, the HVC$_X$ from each juvenile bird showed far more variability. This was particularly compelling comparing the data from the juvenile sibling pair with those from adult siblings. There was a substantial variation in the trains of spikes emitted by different cells within each of the two juvenile siblings (Fig. 6d), far greater variation than seen in adult siblings (Fig. 6a), and these differences were highly statistically significant (see Methods, Additional statistical analysis, Fig. 6d). The neurons from the two juvenile siblings exhibited considerable variation in spike waveforms (Fig. 6e, left and middle panels). There was far more variation in spike waveforms within the juvenile siblings than in adult siblings (cf. Fig. 6e, b, right panels), and this difference was highly statistically significant (see Methods, Additional statistical analyses, Fig. 6e). The distribution of spike train characteristics of cells in all four juvenile birds (including the sibling pair) (Supplementary Fig. 10) was far more scattered within each of

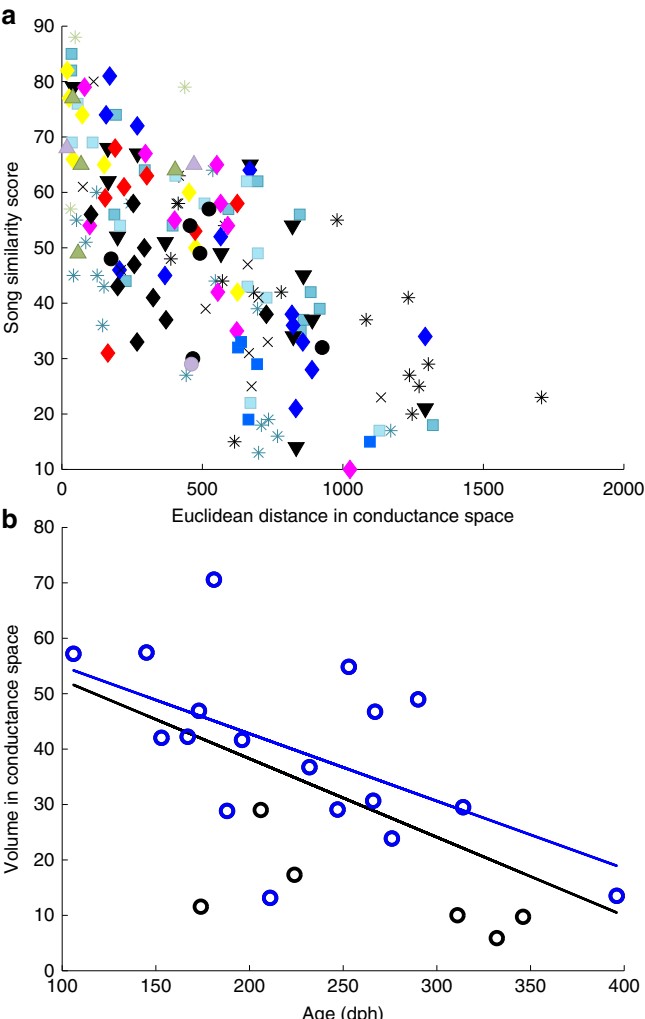

**Fig. 5 Linking intrinsic cellular properties to learning. a** A 2D scatter plot for 18 birds that were modeled and whose songs were recorded. Each bird is represented by a unique color-symbol. For each pair of birds, the similarity score[42] when comparing the songs of the two birds is plotted against the distance between the centroid of conductance values (manual fits) for the two birds. The number of points plotted for a given color-symbol ranges from 1 to 17 (non-identity values in the lower triangular part of the $N \times N$ matrix). **b** A 2D scatter plot showing the five-dimensional conductance space volume for each bird against the bird's age (days post hatch, dph). The blue circles are for birds with $\geq 5$ neurons and the black circles are for birds with 4 neurons or less. (The volume measurement is sensitive to low numbers of recorded neurons, for example note that all black circles are below the regression line.) Both the fit to all birds (black line, $R = -0.57$ and $P = 0.0038$, $t$-test) and to birds with $\geq 5$ neurons (blue line, $R = -0.55$ and $P = 0.016$, $t$-test) are significant (see text), demonstrating age-related changes in the variance of IP.

the birds than the distribution of corresponding values in each of the four adult sibling pairs (compare Supplementary Fig. 8a–c with Supplementary Fig. 10a–c, respectively).

To assess the conductance volumes, since the geometric volume was poorly defined in two of the juvenile birds (with only 5 and 6 HVC_X), we used a "trace" volume measurement (see Methods: Trace volume). Each distribution of points in conductance space for each of the four juvenile birds occupied a larger trace volume (mean $0.355 \pm 0.12$) than observed for adult birds. This difference between the juvenile and adult volumes was highly significant ($P < 10^{-5}$, unpaired $t$ test, $t = 7.9$, d.f. $= 14$, comparing trace

volumes of juveniles with trace volumes of non-manipulated adult birds with $n \geq 6$ HVC_X/bird). There is a hint of a possible age-dependent effect, with the oldest bird (62 dph; Juv3) showing less variation than the three younger birds (Fig. 6f).

A characteristic of the population of juvenile HVC_X neurons was that some but not all neurons exhibited little to no sag in response to hyperpolarizing current pulses, and in these cells termination of the current pulse was not followed by rebound firing (or in some cells was followed with simple rebound depolarization), similar to what has previously been reported for juvenile HVC_X neurons[17]. This was expressed in the HH models of juvenile HVC_X neurons as little to no variation in the $g_{CaT}$ conductance but large variation in the $g_H$ conductance. In general, compared with adult, juveniles exhibited greater variation in the $g_H$ and $g_K$ magnitudes than for $g_{Na}$ or $g_{SK}$. Furthermore, all juvenile neurons were restricted to a small region of the $g_{Na}/g_{SK}$ space occupied by adults (Fig. 6f). Although the sample size is small, this suggests that there are initial genetic constraints on developmental variation in intrinsic properties that vary depending on conductance, and that learning serves to expand some of these constraints (e.g., position in feature space) while restricting others (e.g., variation in feature space). Developmental studies including cross-fostering siblings can help shed light on these constraints. The present results nevertheless indicate that HVC_X intrinsic properties change during development[17,18], not via purely cell autonomous phenomena but in a manner sensitive to network and behavioral constraints.

**Learning rapidly regulates intrinsic properties.** The results suggest a role of auditory feedback in regulating HVCx IP that is first expressed during juvenile song learning and continues into adulthood. To directly evaluate this possibility, we turned to a technique[46] to manipulate the continuous delayed auditory feedback (cDAF) birds experience during singing (see Methods: Delayed auditory feedback). Isolated adult birds (hence singing "undirected" songs) were exposed to between 4 h and 6 days of cDAF (onset of cDAF was in the morning when the house lights were turned on), after which we made in vitro HVC_X recordings from those birds. We analyzed the singing for four birds with shortest cDAF exposure, examining all songs on the baseline day just prior to and on the day of the first experience with cDAF. We observed clear effects of cDAF on singing behavior, preliminary observations that are consistent with more detailed observations made on other birds in an ongoing behavioral experiment. Birds tended to sing fewer songs on the first day of cDAF exposure compared with the preceding baseline day, and each cDAF song tended to be preceded by more introductory notes (Table 1). These effects occurred rapidly, for example they were observed for the bird with only 4 h cDAF exposure. Such rapid change in zebra finch singing behavior has not been reported for other forms of modified auditory feedback which may explain why we found concomitant rapid changes in HVC_X physiology where others had not[31,32].

Interestingly, it was recently reported that a subset of HVC_X in normal birds are active prior to the onset of singing, hence including the interval when birds are producing introductory notes[47]. Here, a compelling result was that the HVC_X of the cDAF adult birds ($N = 9$) showed enormous variation in intrinsic properties compared with non-manipulated adult birds. For example, across the 8 HVC_X recorded in bird White 7 after it experienced 6 days of cDAF there was extensive variation in spike burst patterns, an almost 2× variation in spike frequency, and large variations in spike amplitudes (Fig. 7a; cf. Figs. 1c, 2b, 6a), features never observed in intact adult birds. First spike waveforms in cDAF birds also showed far more variation than

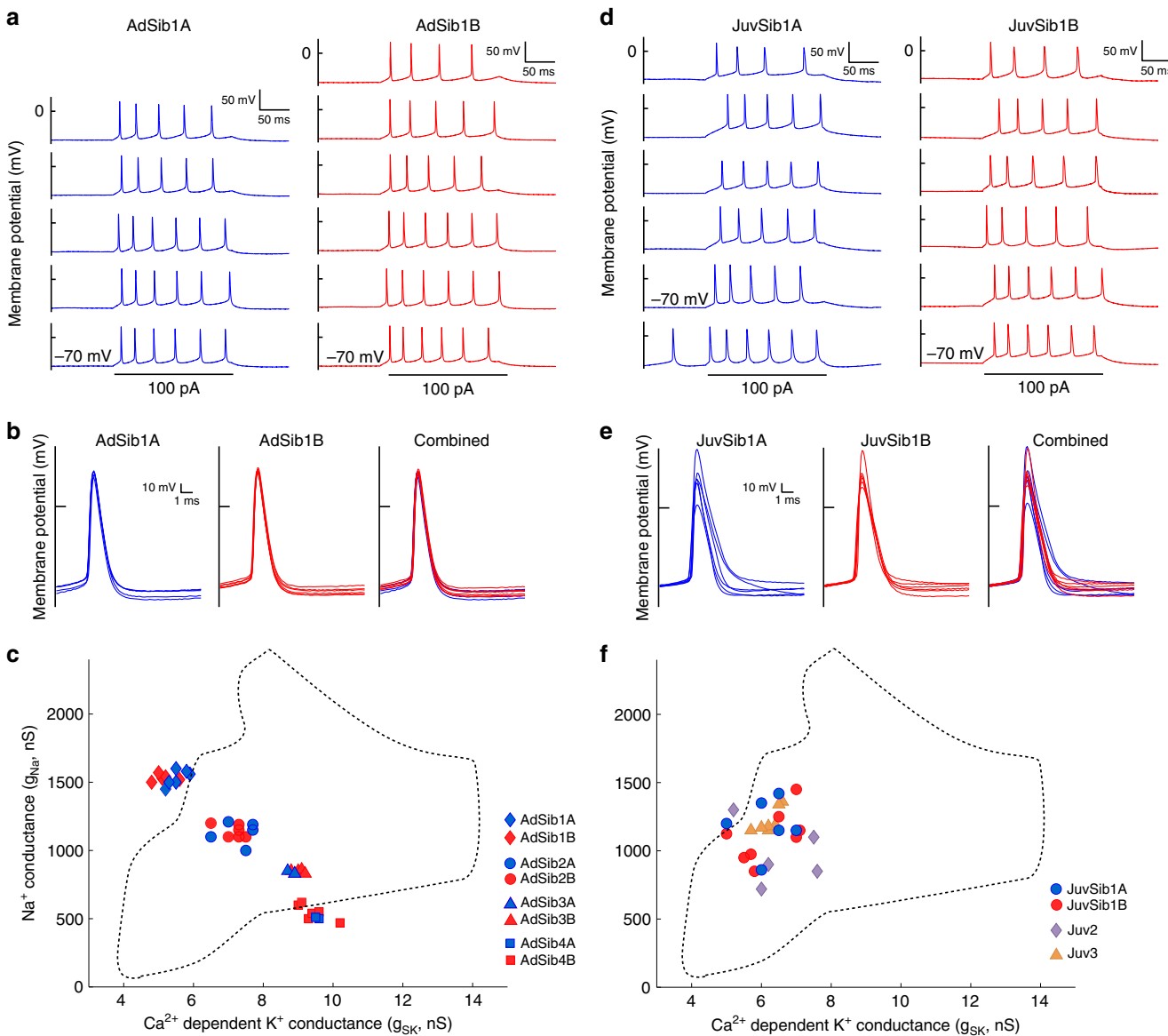

**Fig. 6 A developmental basis for homogeneous intrinsic properties. a** Spike trains in response to canonical depolarizing currents for the neurons recorded from a pair of adult siblings (> 90 dph). Note the similarity of responses within and between the birds. **b** First spike waveforms for each adult sibling (left and middle panels), and then combined (right), showing striking similarity between the siblings. **c** Siblings of each pair occupied similar regions in the parameter space spanned by $g_{Na}$ and $g_{SK}$, while each of the four pairs had non-overlapping distributions with the other pairs. Red and blue differentiate between brothers. The data from these siblings indicates that the parameter space occupied by the species is somewhat larger than indicated in Fig. 3c. **d** Spike trains in response to canonical depolarizing currents for the neurons recorded from a pair of juvenile siblings (JuvSib1A, 57 dph; JuvSib1B, 58 dph). Note the considerable variation within and between the juvenile birds, unlike the adult siblings (cf. Fig. 5a). **e** First spike waveforms from the juvenile siblings, showing considerable variation within and between the birds. **f** Four juveniles occupied an overlapping limited region in the parameter space, with each bird showing considerable scatter in conductance values. Compare with (**c**); both panels share the same X- and Y-axes ranges. For (**c**) and (**f**), the dashed lines show the region (from Fig. 3c) occupied by non-manipulated unrelated adult birds.

**Table 1 Changes in singing behavior for four birds after the onset of continuous DAF exposure.**

| | Orange 168<br>4 h | Pink 89<br>1 day | Orange 116<br>1 day | Orange 148<br>2 days |
|---|---|---|---|---|
| Number of introductory notes preceding each song | *<br>Baseline: 1.76 ± 0.84<br>DAF: 2.43 ± 1.07 | **<br>Baseline: 2.11 ± 1.05<br>DAF: 2.63 ± 1.07 | **<br>Baseline: 2.16 ± 1.28<br>DAF: 2.70 ± 1.11 | **<br>Baseline: 1.68 ± 0.91<br>DAF: 2.06 ± 0.95 |
| Number of songs sung over the interval of analysis | Baseline: 70<br>DAF: 28 | Baseline: 175<br>DAF: 102 | Baseline: 132<br>DAF: 96 | Baseline: 172<br>DAF: 109 |

Numbers of notes analyzed over 1 s preceding each song. Numbers of songs analyzed over first 4 h of the day preceding and the day of DAF exposure for Orange 168, or over the full days for the other birds. Two-tailed unpaired $t$ test. *$P < 0.01$; **$P \leq 0.001$.

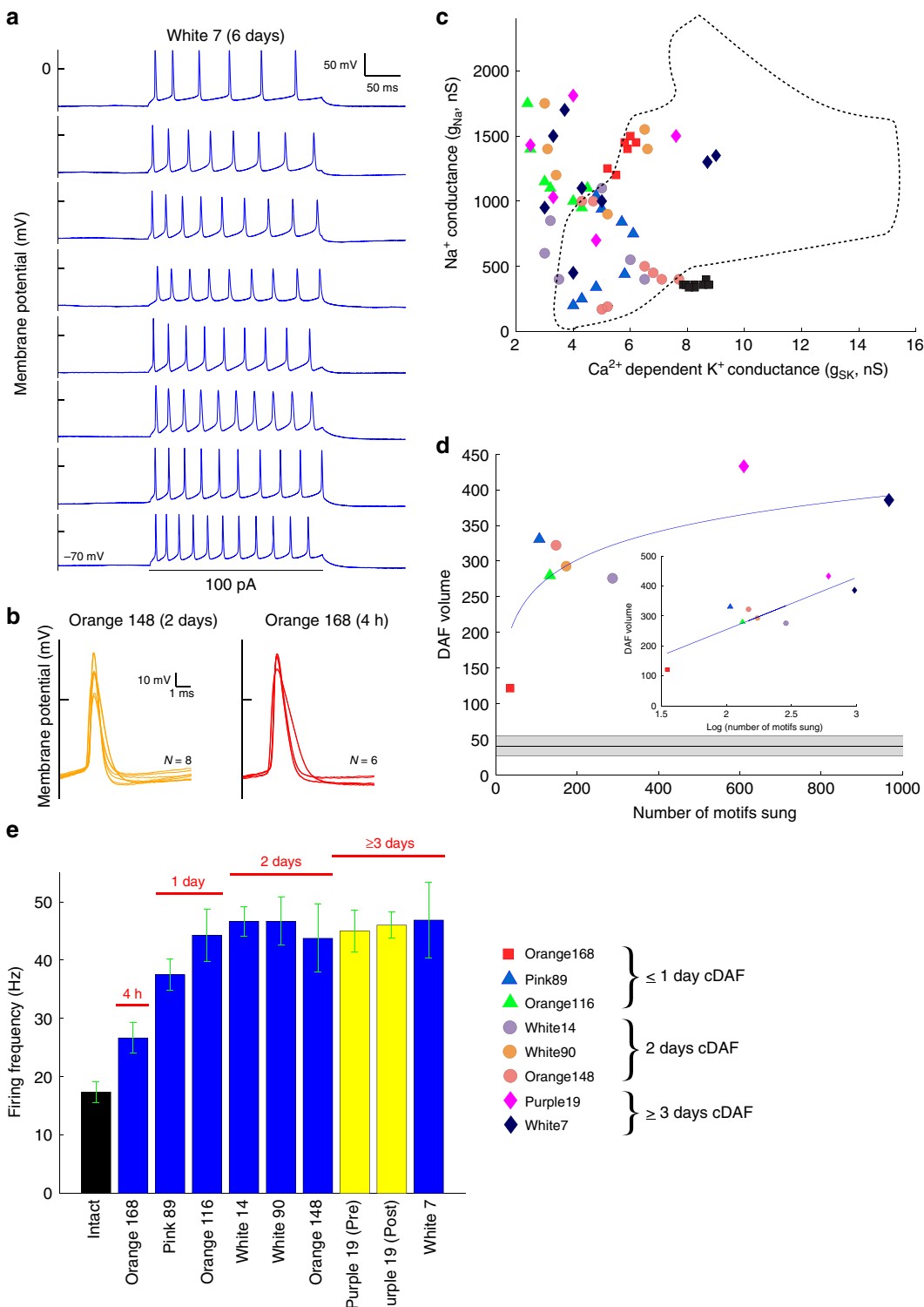

**Fig. 7 Learning rapidly regulates intrinsic properties. a** Spike trains for neurons of bird White 7 (6 days cDAF exposure), showing enormous variation compared with intact birds (e.g., Fig. 1c). **b** Substantial variation in first spike morphologies in a bird after 2 days cDAF exposure; some variation also observed in a bird with only 4 h cDAF exposure. **c** Modeling the neurons in all the cDAF birds showed a loss of tight clustering of ion current magnitudes, and a shift outside the region occupied by unmanipulated adults (dashed line). **d** The trace volume (see Methods: Trace volume) for cDAF birds varied logarithmically with the number of motifs. The inset shows linear-logarithmic plot, demonstrating that all points including the 4 h point fall close to the fitted line. The black line (mean) and gray shaded area (SD) shows the corresponding trace volume for all non-manipulated birds ($N = 17$) with $\geq 5$ HVC$_X$. Legend shows the symbol-colors for birds in panels (**b**–**d**). **e** Mean firing rates. Black bar (intact) is the mean of means ($17.38 \pm 9.02$) for 51 unmanipulated birds, 273 neurons. Each blue bar is one cDAF bird (mean ± SD). The cDAF birds show an increase in excitability on the first day of cDAF exposure and no additional increase in subsequent days. In a bird with 3 days cDAF exposure (Purple 19, yellow bars), the increased spike rate is maintained after bathing the slices with fast synaptic blockers.

expected in non-manipulated adult birds (Fig. 7b; cf. Figs. 2a, b, 6b), and this effect was highly significant (see Methods, Additional statistical analyses, cDAF). (There was no significant change in input resistance between neurons from non-manipulated animals and neurons from animals exposed to cDAF: control, $181 \pm 35$ MΩ, $N = 253$; cDAF, $185 \pm 24$ MΩ, $N = 62$, $t = 0.98$, d.f. $= 313$, $P = 0.15$, unpaired $t$ test) An analysis of the associated ion currents in $HVC_X$ of cDAF birds showed a striking dissolution of the tight clustering of ion current magnitudes seen in non-manipulated birds (Fig. 7c), for example, the conductance geometric volume for White7 was 0.084 of the "species" volume, or 228× the average for non-manipulated birds. Whether these results arise from the abnormal sensory feedback or the motor response to that feedback remains to be determined, but in either case the results demonstrate that intrinsic properties are dynamically regulated in adult animals in a fashion closely tied to ongoing behavior. Future studies can determine if the regulatory response was chaotic or possibly lawful, with the distribution of spike waveform shapes, conductance values, and bursting properties seen in cDAF birds perhaps reflecting the numerous different song syntaxes and variability in other features induced by the DAF exposure[46].

The variation in intrinsic properties was particularly evident for birds with longer cDAF exposure, for example compare the spike waveforms in birds with 4 h of cDAF exposure (Orange 168; Fig. 7b, right panel), 2 days of exposure (Orange 148; Fig. 7b, left panel), and 6 days of exposure (White 7; Fig. 7a). This suggested that the disruption of the clustering was related to the amount of singing the bird had experienced under the conditions of auditory feedback error and was apparent even after only 4 h of cDAF exposure (during which Orange 168 sang 35 motifs, major units of song). A logarithmic relation was observed between the number of motifs each bird had sung and the trace volume of the conductance values for the $HVC_X$ of that bird (Fig. 7d). A linear-logarithmic plot (Fig. 7d, inset) shows that all points including the 4 h point reasonably fit the logarithmic relation. The fit for cDAF birds intersected the largest volume of the 17 non-manipulated birds with $\geq 5$ $HVC_X$ at 5.9 motifs. This estimate is approximate given that the actual relation between conductance changes and cDAF exposure is not likely to be strictly logarithmic, but nevertheless these results indicate that ongoing behavior can regulate intrinsic properties of populations of neurons on very fast time scales, possibly comparable to the time scale at which feedback modifies singing.

We observed that the cDAF birds as a population had smaller $g_{SK}$ values than expected, with many cDAF birds having neurons with lower values than observed for any of the non-manipulated birds (Fig. 7c). Lower magnitude of $g_{SK}$ will manifest as increased excitation, and indeed we observed an increase in excitability of the $HVC_X$ neurons (spike frequency), which was limited to the first day of DAF exposure (Fig. 7e). This suggests a fast mechanism, such as regulation of somatic membrane ion channel density, could be the proximal mechanism giving rise to regulation of intrinsic excitability. In the steady state condition (tested at 3 days after cDAF onset), the increased excitability was maintained after blocking fast synaptic connectivity, indicating that it was intrinsic to the $HVC_X$ (Fig. 7e, Purple 19, yellow bars). Collectively, our data show that $HVC_X$ IP values reflect some experiential signal related to the singing behavior of the birds, tying regulation of intrinsic properties to the deeply investigated birdsong learning model.

## Discussion

A growing body of evidence links modification of IP to learning phenomena[2–4,17,18,48,49]. Here, taking advantage of an animal that produces highly precise and repeatable learned behavior, we describe features of the organization of IP at the level of individual animals. Analysis of raw data traces in response to somatic current injections showed a consistent pattern of homogeneity of $HVC_X$ intrinsic properties within individual adult birds and variation among birds. The homogeneity was maintained by plastic mechanisms. It was developmentally labile[17,18], and in adults a potent form of auditory feedback manipulation[46] resulted in rapid onset and accumulating disruption of IP homogeneity. A model of the Hodgkin–Huxley form incorporating five known $HVC_X$ ion currents[34] did a good job of capturing these results, representing both a concise description of the observations and one hypothesis of the underlying cellular mechanisms. Whatever mechanisms give rise to the observed IP homogeneity, however, notably they are constrained by being related to features of the individual bird's learned song, as shown by the results of the HH modeling. The relation between cellular properties and learning is conceptually significant; it can yield insights into the specifics of bird song learning but also has broader implications for neural coding.

To evaluate the HH model, numerous biophysical parameters other than the principal ion currents were estimated from the literature and held constant, and possible dendritic contributions were folded into a single-compartment model. The limitations of this approach warrants careful evaluation. Variation of the biophysical parameters otherwise fixed in our models[50], as well as additional unknown ion channels that $HVC_X$ might express, and possibly as well similarity/variation in $HVC_X$ anatomy within and among animals, all could yield different results and affect the estimates of the current magnitudes we determined. Techniques from statistical physics applied to problems of data assimilation in high dimensional spaces provide a principled approach to this problem[51,52] and may help to identify if a subset of biophysical parameters are the ones shaped by birdsong learning. However, it is striking that exhaustive testing of a large space of all possible solutions tended to yield a single global minimum that was shared among all $HVC_X$ of a given bird. For example, we reported a large variation (almost 20×) in $g_{Na}$ across animals, a result that warrants follow-up experiments with alternative techniques. This compares with neurons of the lobster stomatogastric ganglion (STG) which can express up to 4× variation in ion current magnitudes in different animals[53]. A comparative analysis within songbird species may help relate differences in singing behavior (e.g., rate, complexity, timing, precision, degree of learning) with variation in cellular properties, and perhaps give insight into variation across taxa.

Our results raise the question of how the IP of the population of $HVC_X$ neurons come to share similar values in a given animal. If homologous neurons with similar morphology receive similar synaptic input (including feedback) then this could tend to regulate similarity of output, in part through mechanisms of IP plasticity. As well as network effects, membrane voltage itself may help to directly regulate coexpression of ion channels thus providing another mechanism through which a neuron's output could affect its own conductance magnitudes[54]. Known features of the functional organization of the zebra finch song system are consistent with robust expression of these mechanisms of IP plasticity. A given zebra finch typically sings a single precisely reproduced song motif, and both $HVC_X$ and $HVC_{RA}$ (the other major class of HVC projection neurons) tend to exhibit notable regularity in the timing of when the very high frequency brief bursts of spikes are emitted during singing[30]. Each spike burst is quite invariant and so seemingly carries relatively limited information beyond when it occurs in song[23,30]. Thus, within a given zebra finch the different $HVC_X$ may receive strikingly similar inputs, have strikingly similar spike bursting activity, and this can

contribute to similar IP. Different $HVC_X$ that are active during singing, however, will emit between one to four spike bursts per song motif[30], yet we detected homogeneity of $HVC_X$ IP in our samples. Unless our sampling in slice unknowingly biased recordings to a subset of $HVC_X$, these considerations seemingly suggest that IP regulation occurs at the level of individual spike bursts.

In toto, these are extreme features of neuronal organization expressed in a "champion" vertebrate model for precision of motor behavior. This suggests that IP is similarly regulated if harder to detect in other species.

The same account can help to explain the relation between IP and a bird's singing behavior. Although it is well established that spike timing during singing is highly regulated in HVC[23], there remains debate as to what features of spike timing are represented in HVC during singing[43,55,56]. If, as by some accounts[43,57], the activity of the population of HVC neurons during singing reflects features of the individual bird's song, then this could serve to sculpt IP of HVC neurons linked to the acoustics of each given bird's song. It remains untested if the timing features of $HVC_X$ during singing vary in a systematic fashion from bird to bird. Here, we used a potent form of abnormal (continuous delayed) auditory feedback which induced rapid effects on singing behavior and $HVC_X$ IP. It would be valuable to assess $HVC_X$ suprathreshold activity under such conditions.

$HVC_X$ IP are also regulated over longer time scales. A recent study[17] demonstrates that $HVC_X$ sag and rebound properties show developmental changes and variability during development. We confirmed those results and extended them to include other features of IP, spike waveforms and excitability (numbers of emitted spikes). In a subsequent study it was reported that untutored juvenile birds exhibited properties similar to adults, but that tutored juvenile birds differed from both, a seemingly paradoxical result[18]. We too observed differences between tutored juvenile birds and adults (Fig. 6f), and our results may help resolve the reported paradox. Given that the prior studies focused on recording from multiple classes of HVC neurons, only few cells were recorded per cell class. This precluded sampling sufficient $HVC_X$ to detect the homogeneity of intrinsic properties within individual adult birds as reported here and left unresolved the distribution of $HVC_X$ IP in individual untutored juvenile birds. If prior to tutoring, $HVC_X$ intrinsic properties are broadly dispersed with each juvenile, this would not have been detected given the sample sizes of 1–3 $HVC_X$/bird[18], and small samples of neurons per bird in untutored and adult birds would both appear broadly and similarly dispersed, resolving the paradox. Future studies exploring variability of intrinsic properties within untutored and tutored juvenile birds may be a compelling approach to elucidating genetic and epigenetic features of developmental song learning.

What is the functional significance of the population homogeneity of $HVC_X$ IP? If all the $HVC_X$ share a common value of IP, then this represents a common baseline from which to detect small perturbations. Small variation in the HVC network could induce changes in the IP of a small subset of $HVC_X$, perhaps manifest as changes in the timing of spike bursts. A given downstream Area X neuron could be sensitive to such changes if it compares spike bursts—perhaps with appropriate delays—received from different $HVC_X$. The same logic extends to other, less homogeneous networks, where neurons with similar IP will tend to more reliably reflect similar outputs for similar inputs. This suggests that stability of neuronal networks includes regularization of IP, with all neurons of a given functional network or neuronal ensemble perhaps tending to share common values of IP. Such regularization would arise from dynamic processes, including learning.

The regularization of intrinsic properties for neuronal ensembles that we report here could also manifest in synaptic organization, which for technical reasons is harder to assess. Yet, we speculate that purely synaptic mechanisms are insufficient to explain information storage in the HVC neural network. We demonstrate that plasticity of the intrinsic excitability of (at least) $HVC_X$ neurons contributes to the song memory engram, presumably using mechanisms known to link numerous classes of ion channels to intrinsic plasticity[58]. The variable dynamics adjusting different ion current magnitudes during learning implies an enduring memory of the prior state, one that is distinct from memories expressed at synapses and having different functional consequences, as has long been postulated[59]. Furthermore, our results (especially for spike threshold) suggest a somatic component of the plasticity of intrinsic properties that we studied. Computationally, in artificial neural networks this implies that the activation function of a model neuron's output is itself subject to learning rules. Biologically, this can serve to integrate across synaptic inputs and/or change output properties of the neuron[60], adding a necessary computational element that was not envisioned in synaptic plasticity accounts of learning and memory phenomena[61].

## Methods

We complied with all relevant ethical regulations for animal testing and research. All animal procedures were approved by the University of Chicago Institutional Animal Care and Use Committee (IACUC).

**Slice preparation**. Slices were prepared from male zebra finches ($N = 76$ birds, yielding 370 neurons) bred at the University of Chicago: 49 non-manipulated adults (> 90 days post hatch (dph)) yielding 221 $HVC_X$, 8 adults (4 pairs) for sibling studies yielding 38 $HVC_X$, 4 adults for pharmacological manipulations yielding 14 $HVC_X$, 8 adults for cDAF experiments (including one with pharmacological manipulations) yielding 62 $HVC_X$, 3 adults for voltage-clamp studies yielding 9 $HVC_X$, and 4 juveniles (55–62 dph) for developmental studies yielding 26 $HVC_X$. Except for the siblings, all the birds were chosen at random. A subset of the birds ($N = 33$) were anesthetized by an overdose of pentobarbital sodium and phenytoin sodium then perfused transcardially with oxygenated standard artificial cerebrospinal fluid (ACSF) which contained (in mM): 119 NaCl, 2.5 KCl, 1.3 $MgCl_2$, 2.5 $CaCl_2$, 1.0 $NaH_2PO_4$, 26.2 $NaHCO_3$, and 22 glucose (osmolarity 285–295 mosM, ph 7.2–7.3); the other 44 birds were anesthetized by isoflurane and not perfused. Both solutions were prepared to the same standards, and since we noticed no apparent difference in the two groups, the data was combined across all birds for purposes of statistical analyses. Birds were decapitated, and the brains were placed in ice-cold ACSF (pregassed with 95% $O_2$−5% $CO_2$) prepared with equimolar sucrose partially replacing NaCl[62]: 72 sucrose, 83 NaCl, 3.3 $MgCl_2$, 0.5 $CaCl_2$, 1.0 $NaH_2PO_4$, 26.2 $NaHCO_3$, and 22 glucose (osmolarity 285–295 mosM, ph 7.2–7.3). Parasagittal (or horizontal) slices were cut (175–400-μm thick, Vibratome 1000) from both hemispheres and placed in an incubation chamber containing ACSF at 37 °C and then allowed to return to room temperature before use. Slices were incubated for at least 1 h in the standard NaCl solution (with no sucrose). In two birds, slices were recorded while being perfused with ACSF warmed to 37 °C. We included these birds in analyzing distributions of spike burst and spike morphological features (Fig. 2) but did not analyze their conductance values or include them in any subsequent analysis.

**Whole-cell recordings**. Slices were superfused with pregassed standard ACSF (2.5–3.0 mL/min) in a submersion chamber. HVC was visible as a dark region when transilluminated. Visually guided recordings were made using unpolished patch pipettes (3–8 MΩ) filled with (in mM) 100 K-gluconate, 5 $MgCl_2$, 10 EGTA, 2 $Na_2$-ATP, 0.3 $Na_3$-GTP, 40 HEPES and 0.5% biocytin, pH adjusted to 7.2–7.3 with HCl. Tight-seal (>1 GΩ), whole-cell recordings were made using a Multiclamp 700B (Axon Instruments), with an active bridge circuit for passing current, 10 kHz low-pass Bessel filter. Series resistance ($R_s$) was estimated according to $R_s = 10$ mV/$I$, where $I$ was the peak current evoked by a 10 mV test pulse. During recordings, $R_s$ was between 3.5 and 10 MΩ before electronic compensation. Only cells that met all of the following criteria were analyzed: a gigaOhm initial seal with a smooth rupture, initial membrane potential ≤−65 mV, a stable membrane potential throughout the recording, and consistent spiking in response to positive current injection. Voltage traces were corrected for an empirically measured liquid junction potential (5−6 mV for standard ACSF and pipette solutions).

Most recordings were at room temperature. In one voltage-clamp experiment, we heated the bath to 37 °C. Data collection was controlled by custom software written in MATLAB. The great majority of the data were collected between 24.5

and 25.5 °C (range 23–26 °C). For two of the five modeled conductances ($I_K$ and $I_h$), the correlation between conductance and temperature is significant, but for all five conductances the relationships are noisy. The clustering of physiological properties on a per-bird basis is apparent for all conductances across all temperatures. For example, for all pairs of birds, comparing the difference in conductance space centroid positions as a function of the absolute temperature difference was not significant ($R^2 = 0.079$, $P = 0.39$). A technical error limits this analysis to all 68 neurons in 16 birds for which we have exact temperature data. The neurons and birds in this limited data set are distributed across almost the entirety of the range of estimated ion current magnitudes for the full data set.

After the formation of a GΩ seal and achieving whole-cell access, a waiting period of 5–10 min followed while the cell was dialyzed with the pipette solution. A standard stimulation protocol was applied to each neuron, comprising 40 pA steps that were either depolarizing (up to 200 pA) or hyperpolarizing (down to −200 pA). Each 200 ms current step was preceded and followed by 200 ms of quiescence. We also applied currents with mathematically defined complex, chaotic dynamics[52] resulting in a random mixture of ramps, sinusoids, steps and other forms. Input resistance was measured periodically during each recording and calculated from the change in membrane potential evoked by small (5–25 pA) hyperpolarizing current steps, 200 or 300 ms in duration.

**Histology**. We filled neurons with 0.5% Biocytin. Slices were processed using standard immunohistochemistry procedures. Mounted sections were screened for successful labeling under epifluorescence using a Zeiss LSM 710 confocal microscope. Images stacks were acquired with Zen2010 software (Supplementary Fig. 1a) and viewed using ImageJ. For the developmental studies, we did not see clear axons for 2 neurons out of the 26 recorded from juvenile birds, and since physiological identification of HVC$_X$ in juvenile birds was far less secure than in adult birds, we excluded these neurons from our developmental analysis.

**Retrograde tracers**. Birds were deeply anesthetized with isoflurane and the head was secured in a stereotaxic frame. Small craniotomies over left and right Area X were made using predetermined coordinates. Tracer (tetramethylrhodamine dextran; Invitrogen; ~2–5 nl) was delivered into Area X bilaterally via a glass micropipette attached to a gas pressure injection system (Picospritzer, Parker). After a 6–7 day period for tracer transport, the birds were used for brain slice electrophysiology studies. HVC neurons that were retrogradely labeled were easily identified in the recording chamber with the use of epifluorescence illumination[34]. Unlabeled neurons were also recorded to confirm that only retrogradely labeled cells display the physiological phenotype of the targeted HVC$_X$ neurons.

**Pharmacological manipulations**. Pharmacological reagents were bath applied using a gravity perfusion system. Recordings were analyzed only for cells with stable $R_s$ values in both control and drug solutions.

We blocked synaptic currents using a cocktail of antagonists of N-methyl-D-aspartate (NMDA), AMPA/kainate, GABA$_A$ and GABA$_B$ receptors: 6-cyano-7-nitroquinoxaline-2,3-dione (CNQX, 10 μM; no. 0190, Tocris), (RS)-3-(2-Carboxypiperazin-4-yl)-propyl-1-phosphonic acid (CPP, 10 μM; no. 0173, Tocris), (3-Aminopropyl)(diethoxymethyl) phosphinic acid (CGP 35348, 50 μM; no. 123690-79-9, Santa Cruz). For blocking GABA$_A$ receptors, we used picrotoxin (PTX, 50 μM; no. 1128, Tocris) in one experiment and SR 95531 Hydrobromide (Gabazine, 10 μM; no. 104104-50-9, Santa Cruz) in the rest. The drugs CNQX and PTX were dissolved in DMSO (final concentration 0.1%), whereas CPP, CGP 35348 and Gabazine were dissolved in water.

In voltage clamp experiments we used, in addition to synaptic antagonists, the channel antagonists: tetraethylammonium chloride (TEA, 25 mM; no. T2265, Sigma), cadmium chloride (300 μM; no. 10108-64-2, Sigma), cesium chloride (5 mM; no. 4739, Tocris), and tetrodotoxin (TTX, 5 μM; no. BML-NA120-0001, Enzo). In one experiment, we used apamin (20–30 nM; no. A1289, Sigma), nifedipine (10 μM; no. 1075, Tocris), mibefradil (5 μM; no. 116666-63-8, Santa Cruz), ZD 7288 (30 μM; no. 1000, Tocris), TEA, and TTX. The drugs ZD 7288, TEA, cesium chloride, cadmium chloride and mibefradil were dissolved in water, apamin was dissolved in 0.05 M acetic acid, TTX was dissolved in 0.1 M acetic acid, and nifedipine was dissolved in DMSO (final concentration 0.1%). Due to the large number of toxic drugs applied to the slice, we lost 48% (8/17) of neurons during the experiments and report only those that remained healthy throughout all drug treatments. The holding potential in voltage-clamp mode was set to −70 mV, and then stepped 10 mV.

**MANOVA analysis**. We performed MANOVA hypothesis testing on spike timing features (Fig. 2) using four covariates from this data set: frequency of spiking, first interspike interval (ISI), second ISI, and third ISI (cells with fewer than four spikes were removed, resulting in $N = 33$ birds). The MANOVA model used was as follows:

$$X_i^b = \mu + \tau_b + \epsilon_i^b$$

for $b \in \{1, \ldots, N\}$ and $i \in \{1, \ldots, n_b\}$ where $N$ is the total number of birds used

and $n_b$ is the number of measurements for bird $b$. We assume here that

$$\epsilon_i^b \sim N_3(0, \Sigma)$$

$\mu$ is the overall mean response for the entire bird population, and $\tau_b$ is the bird-specific effect with the identifiability constraint that $\Sigma_b n_b \tau_b = 0$.

To test the hypothesis that the birds have different mean vectors of covariates, one stringent null hypothesis is that they all have the same mean. This is equivalent to:

$$H_0 = \tau_1 = \tau_2 = \ldots = \tau_N = 0$$

The model above suggests the following decomposition of a single observation as:

$$x_i^b = \bar{x} + (\bar{x}^b - \bar{x}) + (\bar{x}_i^b - \bar{x}^b)$$

where $\bar{x}$ is overall sample mean, $(\bar{x}^b - \bar{x})$ is the estimated bird-specific effect, and $(\bar{x}_i^b - \bar{x}^b)$ is the residual.

The total sum of squares and products (unnormalized covariance) can be decomposed as follows:

$$T = \sum_{b=1}^{N} \sum_{j=1}^{n_b} (x_i^b - \bar{x})(x_i^b - \bar{x})' = \sum_{b=1}^{N} n_b (\bar{x}_b - \bar{x})(\bar{x}_b - \bar{x})' + \sum_{b=1}^{N} \sum_{j=1}^{n_b} (x_i^b - \bar{x}_b)(x_i^b - \bar{x}_b)'$$

If we let $B = \sum_{b=1}^{N} n_b (\bar{x}_b - \bar{x})(\bar{x}_b - \bar{x})'$ representing the between sum of squares and $W = \sum_{b=1}^{N} \sum_{j=1}^{n_b} (x_i^b - \bar{x}_b)(x_i^b - \bar{x}_b)'$ representing the within sum of squares, then $T = B + W$.

Under the null hypothesis, $B = W$. If the bird-specific effects $\tau_i$ were non-zero, this would lead to inflation of $B$. The test statistic is therefore based on "comparing" these two matrices via $W^{-1}B$.

A common test statistic is Wilks' lambda:

$$A = \frac{|W|}{|W + B|} = \frac{1}{|I + W^{-1}B|}$$

which depends on the data only through the eigenvalues of $W^{-1}B$. We reject the null hypothesis for small values of $A$.

The MANOVA null hypothesis above is too stringent in the sense that it is rejected if the mean for any single bird is different than the rest. A more biologically relevant test compares all pairwise differences between birds. To this end, we then performed pairwise MANOVAs on the four covariates while controlling the Type-I error with the Bonferonni correction.

**Additional statistical analyses**. All $t$ tests reported in the main text and Methods were two-tailed, and unpaired comparisons unless otherwise noted.

Figure 2c Clustering in the spike threshold/spike amplitude space. For each neuron in each bird with $N > 2$ neurons, we calculated the distance (in the 2D space of spike amplitudes and spike thresholds) to another randomly chosen neuron for that same bird and for a randomly chosen neuron in another bird. Considering all 241 neurons, the same-bird distances were smaller than the across-bird distances, and the result was highly significant ($t = -18.9$, df = 240, $p = 1.5 \times 10^{-49}$ unpaired $t$ test). This confirms the visual impression that neurons from the same bird were systematically closer to each other (clustered) in the space compared with neurons from other birds.

Supplemental Fig. 3: Clustering in the plateau amplitude/spike amplitude space. The equivalent analysis as for Fig. 2c (above). Considering all 241 neurons, the same-bird distances were smaller than the across-bird distances, and the result was significant ($t = -18.8$, df = 240, $p = 3.5 \times 10^{-49}$ unpaired $t$ test), confirming the visual impression as per logic above.

Figure 6d: Comparing spike trains in sibling juvenile and sibling adult birds. For each sibling bird, we separated the cells into groups by the number of spikes emitted. Any group with only one cell was discarded. For all cells in each remaining group, we computed all pairwise ratios of times (relative to the start of the stimulus) comparing first spikes to each other, second spikes to each other, etc. and accumulated these values across all groups for each bird. We accumulated the distribution of spike time ratios (first spike i/first spike j, second spike i/second spike j, etc.) across each of the two juvenile siblings then compared each juvenile against each of six adult siblings (eliminating the two adult siblings with only two cells per bird). All 12 comparisons were highly significantly different, with the two-sample $F$ test statistic ranging from $F_{(29,23)} = 59.3$ to $F_{(20,23)} = 3.3 \times 10^3$, and $P = 2.0 \times 10^{-15}$ or smaller. This demonstrates that the timing of spike trains in juvenile birds was consistently more variable than the timing of spike trains in adult birds.

Figure 6e: Comparing spike waveforms in sibling juvenile and sibling adult birds. For the two juvenile siblings (six cells recorded in each bird), we found the mean squared difference between one cell's first spike and the first spikes of all cells in the other bird, yielding 36 values. We then did the equivalent analysis for all four pairs of adult siblings. Comparing the distribution of mean squared difference values from the juvenile pair with each of the adult pair, the two-sample $F$ test statistic ranged from $F_{(29,35)} = 0.0017$ to $F_{(9,35)} = 3.6 \times 10^{-4}$, with the largest $P < 8 \times 10^{-12}$. In contrast, 3 of 6 adult siblings were not significantly different from each other, and for the other three comparisons $P = 0.0015$ (two-sample $F$ test) was

the smallest value. This supports the visual impression that the variation in spike waveforms among juveniles was much larger than among adults.

cDAF: Spike waveforms in cDAF birds have much more variation than in non-manipulated adults. Following the logic for Fig. 6e, within each of all eight normal adult birds with ≥8 cells per bird, we computed mean squared spike waveform differences between all pairs of first spikes. These were all pooled together as the variation expected for non-manipulated birds. We then did the equivalent analysis on each of the seven adult DAF birds. Comparing the pooled normal adult data against each of the DAF birds using a two-sample $F$ test yielded $P < 2 \times 10^{-59}$ or smaller. Comparing the pooled normal adult and pooled adult DAF data yielded $F$ $(242,138) = 0.0022$, $P = 2.9 \times 10^{-245}$.

**HH model fitting**. We used a single-compartment conductance-based Hodgkin–Huxley-like biophysical model of HVC$_X$ neurons[34] to estimate the magnitude of the ionic currents. The model includes the spike producing currents ($I_K$ and $I_{Na}$), high-threshold L-type Ca$^{2+}$ current ($I_{Ca-L}$), low-threshold T-type Ca$^{2+}$ current ($I_{Ca-T}$), a small-conductance Ca$^{2+}$-activated K$^+$ current ($I_{SK}$), an A-type K$^+$ current ($I_A$), a hyperpolarization-activated cation current ($I_h$), and a leak current ($I_L$). The membrane potential of each HVC$_X$ neuron thus obeys the following membrane potential balance equation:

$$C_m \frac{dV}{dt} = -I_L - I_K - I_{Na} - I_{Ca-L} - I_{Ca-T} - I_A - I_{SK} - I_h + I_{app}$$

where $I_{app}$ represents the constant DC current injected and $C_m$ is the membrane capacitance.

Manual adjustment of five parameters ($g_{Na}$, $g_K$, $g_{SK}$, $g_H$, and $g_{Ca-T}$) was performed to qualitatively reproduce membrane potential trajectories in response to applied step currents[34]. We fixed the value of $g_{Ca-L}$ because we could achieve the same accuracy of fitting by varying $g_{SK}$, and so could not distinguish between the two. We fixed the value of $g_A$ because in HVC$_X$ it is a very small conductance and has weak effects on the firing pattern[34]. Manual fitting was done blind to the fit of other cells from any given animal. All parameters for model HVC$_X$ neurons were kept the same as in Daou et al.[34] with the exception of the capacitance ($C_m$), which was set to 50 pA in adult HVC$_X$ and 75 pA in juvenile HVC$_X$ (the latter was incremented by 50% due to the fact the juvenile HVC neurons are bigger in size).

For each modeled neuron, parameters were fit by iterative manual tuning for a depolarizing step of 100 pA and a hyperpolarizing step of −140 pA. A fit was considered "good" if the model voltage trace matched the spike frequency, timing of individual spikes, spike amplitude, and resting membrane potential of the biological voltage trace. Critically, the validity of the model was confirmed only if it made good predictions when presented with data sets other than the one it was fit to (e.g., Fig. 3a; see text).

**Rundown**. We performed further calculations to control for any unspecified variation that might emerge in the intrinsic properties of the recorded neurons within a bird as a function of the recording time relative to when the slices were prepared on the day of the experiment. If there is any such time-dependent variation in IP values over the duration of an experiment, then it should manifest in the spike morphological data we were explicitly analyzing in the first cell, second cell, and so on. To evaluate this, we examined data from the 42 normal adult birds (that is, not sibs, not juveniles, not pharmacologically manipulated, not involved in feedback experiments, etc.). We measure the spike amplitude for the first spike of each cell, then computed the ratio of first spike heights (another cell/first cell). If there was rundown, then on average that ratio should systematically vary from 1.0. Considering $N = 179$ such ratios, the average was 0.999903. This was not statistically significantly different from 1.0 ($t = 0.1337$, df = 178, $P = 0.89$ unpaired $t$ test). Also, there was no sudden rundown late in each experiment, because when we considered the ratio of first spike heights (last cell/first cell) for $N = 42$ birds, the average was 0.999914. This was not statistically significantly different from 1.0 ($t = 0.0588$, df = 41, $P = 0.95$ unpaired $t$ test). The results reject the hypothesis that a principal measure of spike morphology derived directly from the raw data (spike amplitude), which should be related to max $g_{Na}$ and max $g_K$, varied systematically (or hardly at all) across experiments within each bird.

We also checked whether there was any variation over the time course of days. In particular, within the group of 42 birds, we identified 12 experiments conducted 1 day apart ($D = 1$) and 10 experiments conducted 14 days apart ($D = 14$). For each bird, we calculated an average of all first spike amplitudes (as per Fig. 3b), then calculated the ratio (smaller average spike amplitude/larger average spike amplitude). (We added the correction to get all values ≤1.0 because there is no confidence whether the hypothesized variation across days should increase or decrease IP values. This was an attempt to increase the chance of arriving at a statistically significant difference.) Comparing the distributions for the $D = 1$ with the $D = 14$ groups of birds yielded a difference that was not statistically significant ($t = 0.1823$, df = 20, $P = 0.8571$ unpaired $t$ test). This rejects the hypothesis that there were systematic differences in the results depending on intervals of days or more.

Finally, we compared the $D = 1$ and $D = 14$ groups of birds with the 4 pairs of sibling birds: one pair of sib experiments conducted 1 day apart, two conducted 2 days apart, and one conducted 4 days apart. In both cases the differences between the two groups were significant ($D = 1$ and sibs: $t = 2.4664$, df = 14, $P = 0.027$ unpaired $t$ test; $D = 14$ and sibs: $t = 2.1408$, df = 12, $P = 0.054$ unpaired $t$ test).

This is remarkable given the very small sample sizes (4 pairs of sibs, 10 or 12 pairs of other birds). Thus, the difference between pairs of sibs recorded on different days and pairs of randomly chosen birds recorded on different days is compelling. All these analyses support the conclusion that our experiments were well controlled with regard to unspecified variation in the recordings.

**Model predictions and cross-validation**. To quantify the validity of the goodness of fit for the manually-adjusted models we developed an error function that attempted to capture the heuristics used for manual fitting. The error function used mean squares to compute the difference between the biological voltage trace and the model simulation by adding the individual differences between the two for each of the following features: plateau amplitude, spike amplitude, spike duration, number of spikes, the timing of each individual spike, resting membrane potential, sag ratio and rebound firing. These features were only well defined for the responses to step current pulses, hence we did not estimate goodness of fit for the responses to chaotic current pulses.

The ratio of the error of the prediction for non-fitted current injections, relative to fitted 100 pA injections was $1.53 \pm 0.35$ (150 pA injections), $1.54 \pm 0.24$ (75 pA injections), and $1.69 \pm 0.47$ (175 pA injections) ($N = 151$ HVC$_X$, 38 birds). This indicates good predictions across the population of neurons. Comparable predictions were observed for 75 and 150 pA injections (paired $t = 1.59$, d.f. = 150, $P = 0.11$) but predictions for 175 pA were significantly worse (paired $t = 3.30$, d.f. = 150, $P = 0.0012$, comparing 175 and 150 pA predictions). At 175 pA, the model was more excitatory than the neurons, perhaps indicating some inaccuracy in the curves for activation dynamics that we were using for the various ionic currents.

To assess the goodness of prediction we conducted a cross-validation analysis. First, we picked the neuron with the best fit in each of 38 modeled birds, thereby eliminating pseudo-replication within birds and reducing the size of the data set. Then we computed $(38-1)^2$ errors (avoiding duplicate calculations), each a comparison of a neuron's model prediction (for 150 pA) to the actual response of all the other 37 neurons for 150 pA. Finally, we normalized these values, taking the ratio of each cross-comparison error to the corresponding error for neuron's actual fit (for 100 pA). This measure was applied only to responses to step current pulses and not to responses to chaotic current pulses. Good fits implied that the in vitro responses we were measuring in slice were dominated by a single compartment, and that variation of the five principal ion conductances was sufficient to predict variation in responses exhibited by the actual neurons. We note that whereas network inhibition and excitation could contribute to the total current we measured for a neuron, the pharmacological manipulations in this study indicate that synaptic currents have little to no effect on spike morphology, which was used in the fitting process.

The cross-validation analysis rejected the hypothesis that the prediction for 150 nA current injections was no better for the actual cell's response than for the response of HVC$_X$ of other birds (Supplemental Fig. 5). The mean of the cross-validated error values was 4.9× greater the mean for the actual error values, and no cross-validated predictions for a given neuron were smaller than values for that neuron's actual model prediction. While the null hypothesis is weak, nevertheless rejecting it demonstrates that the modeling could capture variation in responses of HVC$_X$ of different birds.

**Exhaustive parameter searches**. We used a supercomputer (Cray XE-6, 24k cores) to conduct exhaustive parameter searches on the space of ion current magnitudes of 70 HVC$_X$ in 12 birds (54 neurons from 10 intact birds, 16 neurons from 2 DAF birds). $g_{Na}$ ranged between 200 and 2500 (step size of 20), $g_K$ ranged between 50 and 750 (step size of 10), $g_{SK}$ between 0 and 15 (step size of 0.2), $g_H$ between 0 and 15 (step size of 0.2), and $g_{Ca-T}$ between 0 and 10 (step size of 0.2). We used the error function described above (see the 'Model predictions and cross-validation' section) to compare model and biological traces. Therefore, for each of the 70 neurons, we examined $2.43 \times 10^9$ possible solutions spanning the 5D space. Any model simulation that generated a spike train exhibiting different numbers of spikes than the biological trace was disregarded prior to computing the error.

Examining slices through the six-dimensional space of ion current magnitudes vs. model error revealed one or multiple local minima, with a single global minimum (best model) for that neuron residing in a distinct concavity (Fig. 3b). Examining the current magnitudes that varied the most in the population ($g_{Na}$, then $g_{SK}$), in 58 neurons, by visual inspection the manual fit was relatively close to and resided within the same concavity of the global minimum (e.g., Fig. 3b), with the global minima for neurons from the same bird clustered together albeit less tightly than for manual fits (Supplementary Fig. 4). For the remaining 12 neurons (2 birds), the global minimum was distinct from the values achieved by manual fitting. Even in these cases, the values of global minima and patterns of the error functions were similar for the HVC$_X$ within each bird and varied across the HVC$_X$ of different birds (Supplementary Fig. 5). Moreover, for all 12 of these neurons the models generated unrealistic spike morphologies (Supplementary Fig. 4), implicating inadequacies in the error function. These limitations notwithstanding, the results rule out that bias in manual fitting significantly contributed to the pattern of variation in ionic current magnitudes within and across birds that we had observed.

The main goal of the parameter searches was to test the validity of the manual fit estimates and the conclusions regarding within-bird homogeneity and across-birds heterogeneity. To this end, we noted that the parameter sets corresponding to error minima that the exhaustive searches converged to showed clear clustering of

conductances (Supplementary Fig. 7), even in cases where the automated searches did not converge to the same values as the manual fits. The discrepancies between the two is largely due to limitations in the way the error function was constructed. On occasion the automated procedures identified global minima whose corresponding voltage traces were not biologically realistic, but exactly fit the duration and height of the initial components of the action potential (Supplementary Fig. 4). The manual fit, however, elicited a more realistic looking trace when compared with the biological recording in terms of its plateau amplitude and the smooth AHP following action potentials. Because the computed error is a summation of the difference between the model simulation and the biological recording from each of the extracted features, the minimum does not always represent the most optimal solution, although it matches one or more of these features better.

**Conductance volumes.** An estimate of the volume $V(B)$ occupied by $N$ HVC$_X$ in bird $B$ was calculated as follows:

$$V(B) = \sqrt{\frac{1}{N-1}\sum_{i=1}^{N}\left\{(\overline{g_{Na}} - g_{Na}^i)^2 + (\overline{g_K} - g_K^i)^2 + (\overline{g_{SK}} - g_{SK}^i)^2 + (\overline{g_{CaT}} - g_{CaT}^i)^2 + (\overline{g_H} - g_H^i)^2\right\}}$$

where $g_X^i$ represents conductance $X$ for neuron $i$, and $\overline{g_X}$ represents the mean of all conductances $X$ for the HVC$_X$ of that bird, that is

$$\overline{g_X} = \frac{1}{N}\sum_{i=1}^{N} g_X^i$$

Note that $V(B)$ is closely related to the sample covariance matrix, which is defined as

$$\sum(B) = \frac{1}{b}\sum_{i=1}^{b}(x_i - \bar{x})(x_i - \bar{x})^T$$

where bird $B$ has b HVC$_X$. The trace of the covariance matrix,

$$\text{trace}\left(\sum(B)\right) = \frac{1}{b}\sum_{i=1}^{b}\|x_i - \bar{x}\|^2 = V(B)^2$$

is the sum of the eigenvalues, which measure the spread of the data along each orientation. For example, the largest eigenvalue measures the variation along the direction where the data varies the most, the second largest eigenvalue measures the variation along the second-most-varying direction, etc. The determinant of the covariance matrix (the product of all eigenvalues) yields a measure of the "volume" in the geometric sense, and we refer to this as the geometric volume. The performance of this measure was good especially for birds with > 6 HVC$_X$. For instance, the volume estimates were relatively stable for different non-manipulated birds. Bootstrap and cross-validation simulations also supported this conclusion.

**Trace volume.** Degeneracies arise when the number of observations is not large compared with the number of dimensions in the data. This can yield near-zero volumes, which are not biologically informative, especially when certain dimensions consistently show larger variations than the others (e.g., $g_{Na}$ and $g_{SK}$). As a result, we modified the definition of volume by summing the variations (summing the eigenvalues, related to the trace of the matrix), and taking square root to rescale. We refer to this as the "trace" volume.

All volumes computed (whether for individual adult non-manipulated birds, juveniles, or cDAF birds) were normalized with respect to the volume of the "species" estimated from all neurons from non-manipulated adult birds. Geometric volumes of individual birds were normalized to the "species" geometric volume. Trace volumes of individual birds were normalized to the "species" trace volume.

**Mahalanobis distance.** Consider the maximal conductances of any bird $G$ as an $n$-by-5 matrix, where $n$ represents the number of neurons and 5 represents the 5 maximal conductances that were varied for each neuron. Then, for any two birds $G_1$ and $G_2$, to compute the Mahalanobis distance, we first computed the mean for each matrix and then took the centered data ($Cen_i$) around the mean as such

$$Cen_i = G_i - \mu(G_i)$$

where $G_i$ is the corresponding matrix for bird $i$ (for $i = 1, 2$), and $\mu(G_i)$ is the mean of the matrix $G_i$.

We then computed the pooled covariance ($P$) of the covariance matrices $C_1$ and $C_2$ for birds $Cen_1$ and $Cen_2$, respectively.

Finally, the distance was defined as

$$D = \sqrt{x.P^{-1}.x'}$$

where

$$x = \mu(G_1) - \mu(G_2)$$

This distance is different than the Euclidean distance in the sense that it is unitless and scale-invariant, and takes into account the correlations of the data set.

**Song recordings.** Birds were isolate housed in sound attenuation booths (AC1, Industrial Acoustics Corp.). Songs were recorded with an omnidirectional microphone (Earthworks SRO) through a preamp (AudioBuddy, M-Audio) and USB interface (Delta 1010, M-Audio), using custom software based on the linux Jack audio toolkit. An exemplar motif was manually selected for the Euclidean/Mahalanobis distance calculations.

**Delayed auditory feedback.** We used an experimental procedure for exposing birds to continuous delayed auditory feedback (cDAF)[46]. Briefly, songs were recorded using custom software based on the linux Jack audio toolkit. High-quality recordings of the songs of eight birds were achieved using piezoelectric accelerometers (BU-1771 or 7135, Knowles Acoustics, IL, USA) that were chronically affixed onto the skulls of the birds. Recording the songs by means of substrate-borne vibrations (bone conduction) permitted airborne playback at high gain of veridical copies of the bird's songs while avoiding uncontrolled positive feedback in the system. A fixed delay of 100 ms was used in these studies. The mixture of a bird's singing and the delayed song were recorded by a microphone and saved to disk along with a recording of the accelerometer signal. Such an environment can lead to immediate dysfluencies in humans and ultimately (over longer intervals) to striking dysfluencies in songbirds[46]. Songs were parsed from long continuous recordings using Sound Analysis Pro from birds experiencing 4 h, 1 day, and 2 days of cDAF. Noise in one of the 1-day birds reduced the reliability of SAP parsing and we abandoned that analysis.

Birds subjected to cDAF were maintained on 12:12 reverse day–night light schedule, typically for a week prior to any surgeries or song recordings. In birds where cDAF exposure was ≥1 day, the bird was captured at the end of the subjective day and slices were immediately prepared for subsequent electrophysiological experiments. The bird with 4 h of cDAF exposure was killed at the end of that interval.

The bird, number of neurons recorded, and total duration of cDAF exposure (12 h/day) was as follows: Orange 168 (6 neurons, 4 h), Pink 89 (8 neurons, 12 h), Orange 116 (7 neurons, 12 h), White 14 (6 neurons, 24 h), White 90 (6 neurons, 24 h), Orange 148 (8 neurons, 24 h), Purple 19 (5 neurons, 36 h), White 7 (8 neurons, 72 h).

**Reporting summary.** Further information on research design is available in the Nature Research Reporting Summary linked to this article.

## Data availability

The data that support the figures within this paper and the other findings of this study are available upon reasonable request. A reporting summary for this Article is available as a Supplementary Information file.

## Code availability

The computer code to produce the figures of this paper and the other findings of this study are available upon reasonable request.

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

## Acknowledgements

Christian Hansel and Jason N. MacLean engaged in valuable conversations as these results developed. J.N.M., Nicolas Brunel, C.H., and C.R. Gallistel provided useful comments on the paper. Peter Malonis developed the code and ran most of the data sets subjected to exhaustive parameter searches. Matt Bonakdarpour, Wanting Xu, and Michael L. Stein provided statistical advice via the Consulting Program in the Dept. Statistics at Univ. Chicago, and carried out the MANOVA, conductance volume, and PCA analyses. Daniel D. Baleckaitis provided valuable technical support. This work was supported in part by resources provided by the Computation Institute and the Biological Sciences Division of University of Chicago and Argonne National Laboratory (NIH 1S10OD018495-01), grants to DM from NIH (DC012859, NS094831) and DoD (BAA 11004985), and a seed grant to DM from the Big Ideas Generator, University of Chicago. DM is a member of the Grossman Institute for Neuroscience, Quantitative Biology and Human Behavior at the University of Chicago.

## Author contributions

A.D. conducted the experiments. Both authors contributed to the experimental design, data analysis, and manuscript preparation.

## Competing interests

The authors declare no competing interests.
