## [Peer Review File · Nature Communications]

Reviewers' Comments:

Reviewer #1:

Remarks to the Author:

In their paper, Daou and Margoliash examine intrinsic properties of basal ganglia-projecting neurons in HVC of the zebra finch. Surprisingly, they find that responses to current steps administered to the cell body are considerably more similar across neurons within one bird than they are across birds. They use synaptic blockers to demonstrate that these responses are due to intrinsic properties of neurons rather than network effects. The homogeneity of intrinsic properties does not appear to exist in juvenile birds, and birds exposed to distorted auditory feedback show differences in their intrinsic properties compared to controls.

Overall, these findings are generally exciting, with the potential to add a major new insight to our understanding of the songbird brain as well as neural networks more broadly. However, the manuscript needs some work. Once the points below have been addressed in a satisfactory manner, I would be supportive of publication in Nature Communications.

MAJOR CONCERNS

(1) The weakest part of the paper is the attempt to model ionic conductances using a Hodgkin-Huxley approach. The attempt this using a single compartment model, despite the fact that an overwhelming amount of literature has found major intrinsic changes of ionic currents in the dendrites (and even single dendritic branches). It is unlikely that these changes will be easily detectable from somatic recordings, calling into question the utility of this modeling effort. For instance, the final paragraph of the discussion points to the importance of 'somatic intrinsic properties' (Page 19), which rings quite hollow in my view. Furthermore, parameters of the model are hand tuned and not verified using other approaches such as mRNA levels of specific channels (e.g., Shultz, Goaillard, and Marder, Nat Neuro, 2006), immunostaining, or any other 'ground truth approach'. The reason that Eve Marder's work was compelling is that a large spectrum of models could explain the pyloric rhythm that was observed. Here only a single hand-tuned solution is presented, and that has the potential to be misleading. Therefore, as it stands, I think that the modeling component represents an unnecessary layer of obfuscation that should be removed wherever possible and replaced with primary observations, such as spike shape, spike rate, etc. For instance, I find Supplementary Figure 10 to be far more convincing than Figs. 6c&d – simply because the scientific point can be made with primary measurements alone rather than having to invoke speculative biophysical conductances.

(2) In the methods, it is stated that 'Most recordings were at room temperature' (Page 20). The authors should clarify this statement. For instance, if different birds were recorded at different temperatures, then this could easily explain the primary phenomenon described.

(3) At several points in the paper, the degree of speculation outpaces the amount of available data. For instance, the 'long-sought physiological signal' is misleading. Although the changes in HVC(X) neurons are intriguing, we don't yet know the impact of these changes. For instance, it is difficult to think of a method to selectively halt the changes in intrinsic properties in those neurons to determine the effect on the network and behavior. Until that time, we are left with a correlative result whose significance (or lack thereof) will be determined at a future point. This does not take anything away from the present finding, but too much speculation may serve to distract from what is otherwise a fascinating empirical observation.

(4) Very recently, a paper was published from Richard Hyson's group showing the impact of tutor song experience on intrinsic properties of HVC neurons (Ross et al., J Neurosci, 2018). Those data are broadly consistent with the work presented here without stealing away any of the major thunder. That manuscript should be cited and thoroughly discussed. Also, given that paper and Hyson's previous work, the authors should say that their data 'confirm' (rather than 'indicate') that intrinsic properties are changing through development (Page 14).

(5) Overall, the manuscript would benefit from careful proofreading in order to clarify the progression and interpretation of experiments and to improve logical flow. For instance, on Page 2, the following

fact are presented: (i) Zebra finches do not modify the basic song motif structure rapidly in response to perturbations, (ii) Zebra finches require auditory feedback to maintain their songs, (iii) Therefore, we need to find the parameter that is regulated. This is not fair. Of course, auditory feedback does not lead to major song changes at the level of motif structure, but conditional auditory feedback (Tumer et al., 2007) can lead to lawful pitch (Andalman et al., 2009) and timing (Ali et al., 2013) shifts in zebra finches that are likely due to spiking changes in LMAN and HVC, respectively. There is no need to create a vacuum that intrinsic property changes must fill.

(6) On Page 3, it is stated that 76% of neurons had a RMP ≤ -70 mV. Rather than provide this number, I would rather see the mean and variance of the RMP for all recorded neurons as well as the range, mean and variance of the current needed to hold neurons at -70 mV.

(7) In Figure 3, the 'plateau amplitudes' for individual neurons are quantified, but the associated raw data are not shown. It would be considerably easier to understand the figure if these traces were included for example neurons.

(8) On Page 5, it was stated that input resistance varied from neuron to neuron 'as expected'. I find this absolutely *unexpected* given the results presented for plateau amplitude, spike rate, and interspike interval – all factors that should absolutely depend on input resistance. The authors should provide a clear explanation for this.

(9) The authors state that they administered hyperpolarizing current traces in addition to their 'canonical' 100pA depolarizing current traces. These hyperpolarizing current traces and associated data (e.g., membrane potential 'sag') should be used to further characterize the intrinsic properties of neurons presented here.

(10) To help address the overall concern about other uncontrolled sources of experimental variance, the authors should examine the relationship between intrinsic properties and the series resistance measurement for each recording. If recording quality has nothing to do with the results obtained, then there should be no relationship.

(11) On Page 10, the authors engage in some analytical acrobatics that leave me a bit concerned. There, they mention that 'one outlier bird gave rise to all the non-significant results', and once those data were removed 'all but one were significant'. I don't see the benefit of curating the data in this way, and the authors should remove this from their analysis.

(12) The estimation of the number of motifs needed during DAF is somewhat flawed. As the authors point out, this relationship is not necessarily logarithmic and appears to be strongly influenced by a single point (Figure 7d). More data are needed to better understand this relationship, and – until that happens – the claims should be softened accordingly. Also, the authors do not quantify how much degradation results from the DAF – how was this related to changes in intrinsic properties?

MINOR CONCERNS

(1) The results relating to run-down should be transferred from the Methods to the Results, specifically the lack of impact of the number of days between experiments or the variation in measured responses throughout the duration of the experiment.

(2) 'IP' is used throughout the text without being defined at any point.

(3) On Page 10 (third line from bottom), the value for R is missing.

(4) On Page 17, the authors speculate that populations of HVC(X) neurons could be related to different syllable types. Given the homogeneity of intrinsic properties, how would these populations be defined?

(5) On Page 17, the authors speculate that 'common total synaptic drive' could induce common values for 'intrinsic properties'. If this is true, then that suggests that HVC neurons whose intrinsic properties are heterogeneous (e.g., juveniles or adults following DAF) have heterogeneous synaptic input? That is a strong – and testable – prediction.

(6) On Page 17, the statement that 'HVC(X) spiking is characterized by hyperpolarization during singing' is confusing: How can spiking be characterized by hyperpolarization?

(7) Debanne has a wonderful new review on plasticity in intrinsic properties (Debanne et al., 2019)

that should be cited here.

(8) The statement that HVC(X) and HVC(RA) neurons project exclusively to X and RA, respectively is not formally true. In fact, in a recent paper from Mike Long's lab, 2 out of 40 projection fully reconstructed HVC projection neurons sent axons to both structures (Benezra et al., JCN, 2018).

Reviewer #2:

Remarks to the Author:

This is an exciting study showing that intrinsic excitability properties in a subtype of neuron in the avian song circuit critical for vocal learning are intricately aligned with the type of song that the bird produces. By measuring a set of 5 ionic currents in these neurons and using modeling to confirm that these currents are sufficient to predict underlying cellular excitability, the authors show convincingly that the ionic conductances that determine a neuron's specific intrinsic excitability are much more similar between neurons in an individual than they are across individuals and that the similarity between individuals is highly correlated with song relatedness. These findings suggest that intrinsic excitability is tightly linked with the neural, and by extension circuit, demands for the songs that individual birds produce.

While this finding is surprising and highly interesting in its own right, one of the most remarkable aspects of the study is the finding that excitability profiles are highly plastic and that sensory perturbation for as little as 4 hours (using continuous delayed auditory feedback) will cause a change in these highly stable excitability profiles. Because the neurons where this change is observed (HVC neurons that project to the basal ganglia) play a critical role in vocal plasticity, these findings imply that intrinsic excitability might be one of the first targets for inducing vocal plasticity following sensory feedback perturbation. The observation that juvenile birds during the song learning stage lack the high degree of homogeneity in intrinsic excitability that is observed in adults, further suggests that the regulation of intrinsic ionic properties is fundamental to the vocal learning process. These are significant findings because they offer a new way of thinking about circuit-level plasticity which tends to be highly focused on changes in synaptic weights. While there is a small but growing literature showing that changes in the expression pattern of specific ionic conductances can be intimately linked to neural plasticity, this study is the first to show the importance of intrinsic excitability and modulation within the context of a complex learned natural behavior.

Overall I found this to be a thorough and well-crafted study that combines experimental and computational approaches. The results section of the paper is clear and easy to follow. My biggest issue with the current manuscript is the discussion section which is difficult to read and generally does not do this study justice. The figures are generally good but could benefit from some minor modifications to make certain points clearer. Although not necessary, in my opinion, for this first description of the phenomenon, it would have been nice if the authors could have used genetic relatedness to further highlight the point that similarity in intrinsic excitability is related to the type of song the bird produces rather than heritability of the trait. In future experiments it would be interesting to compare intrinsic excitability in siblings tutored with different songs. It would also be interesting to compare excitability profiles in adult siblings singing highly similar songs (as in figure 6c) but with one sibling experiencing cDAf while the other is treated to a neutral control condition.

General Comments:

Discussion:

The discussion section would benefit significantly from an overall re-organization. As written it seems

to lose the thread that is so clearly laid out in the results section. The discussion also contains a number of long sentences that feel grammatically awkward and therefore difficult to follow. In general, the discussion seems to jump around quite a bit and is quite heavy on "song system" jargon which will make it difficult for a reader to follow if he/she does not have the appropriate background. The lack of a figure of the neural circuit makes it particularly difficult to follow some of the more complex arguments that the authors lay out in terms of mechanisms that might drive the observed similarity in intrinsic excitability.

Figures:

Aside from adding a "song system" figure that could be used to illustrate some of the arguments made in the discussion section, I am wondering if the authors could come up with a more effective graphical representation to show the changes that occur when young juveniles transition to adulthood. The same suggestion holds for showing the changes that occur following cDAF. Figure 6c&F as well as figure 7c are particularly difficult to visualize especially on the low quality pdf because the low intensity symbols are hardly visible and even if they were, the full meaning of the finding lacks that immediate pop out effect.

Minor comments:

Abstract: I recommend removing the word "if not quicker" in the statement "degraded, within hours if not quicker, in adults singing" given that the authors do not provide any evidence for such claim.

Page 2 – bottom

For the following statement "Furthermore, in response to feedback perturbation during singing HVC spiking activity do not change in zebra finches whereas reliably changes are observed", the authors should also include several references showing that such perturbations do not affect synaptic activity (e.g. Hamaguchi et al. 2014 and Vallentin and Long, 2015). There is also a typo in that sentence.

Page 3 – top

It would be helpful to add a short half sentence here (details can remain in the supplemental section) stating that neurons are retrogradely labeled with tracer following tracer injection in Area X. This might also be a great place a modified figure of the song system. This figure can then serve the dual purpose of highlighting the methods as well as serving as a reference for the various arguments in the discussion.

Figure 2

In figure 2b it would be helpful to indicate that each color represents the average spike waveform for different birds. Would it be possible in this figure to also include some measure of spike waveform variance by including +/- SD contours?

Page 6 – middle

The following definition for the SK current as "small potassium-mediated calcium channels (ISK)" is awkward. It almost reads as if it were a calcium current. Why not describe it simply as a calcium-dependent potassium channel?

Page 8 – top and Figure 3b

This figure is difficult to interpret. The rationale for this figure should be revised in the text and the figure itself should be modified. In particular, it is quite difficult to notice the small arrow and circle on the bottom right of the figure. It is also not clear how to interpret the error axis. What is the significance of an error value of 400? More detail needs to be provided in the legend as well as the text.

Figure 6c

In addition to the difficulty of seeing the shaded symbols (see comment above), the definition of the different birds was a bit confusing to me. What threw me was the sibling pair with different (leg-band I assume) names (Gray15 and White3). Why not simply relabel symbols as (sibling 1 & sibling 2; parent colorX)

Figure 7

The figure legend is too sparse to provide a clear explanation. This is especially true for figure 7c. Figure 7d is also difficult to interpret because the legend indicates that the histogram represents changes on the first day of cDAF exposure, yet several of the data points are from birds that are described in figure 7b to have been exposed to cDAF for two days or more.

Page 16 – bottom

A few examples of confusing and/or grammatically awkward sentences that are rather typical for the discussion section

“Many songbird species exhibit within-individual variability in their adult singing behavior, not only well known cases such as Bengalese finches, canaries, mockingbirds, and starlings, but including at least some species that are commonly thought of as having a single song repertoire (e.g., indigo buntings), and also in response to seasonal changes and other factors.”

“ In zebra finches, song output and song system neurons suprathreshold activity was found to be refractory to acutely altered auditory feedback as well as manipulations of other forms of sensory feedback, albeit cDAF was not tested.”

Reviewer #3:

Remarks to the Author:

This is an ambitious study that addresses important but still largely unexplored questions in birdsong biology. The authors have conducted detailed recordings in slices and modeling efforts to characterize the intrinsic physiological properties of HVCx neurons, a key component of the song control circuitry representing the source of input from the vocal motor pathway into the anterior pathway for vocal learning. They provide substantial evidence that these cells have similar spike waveform properties and spike train timing within individual birds, but also that these properties vary considerably across individuals, apparently in correlation with individual variability in song bioacoustics. The study also provides evidence of higher variability of these parameters in juveniles at an age when their songs are not yet highly stereotyped, and that these parameters seem to change in adults as a result of auditory feedback disruption. These findings are presented as evidence for song and error representation in zebra finches. Overall, the effort described is quite substantial and the study is a potentially important contribution, however some significant weaknesses decrease enthusiasm for the paper as presented,

both in terms of data and interpretation. The main issues are listed below, followed by more specific comments and minor points.

Several statements and conclusions are not backed by rigorous statistical analysis and appear more like qualitative assessments of the data. This includes statements like "spike amplitude and plateau amplitude clearly segregated neurons from different birds"; "the model of a given neuron was far better at predicting responses of that neuron than were models of neurons from other birds"; "there was a substantial variation in the trains of spikes emitted by different cells within each of the two juvenile siblings (Fig. 6d), far greater variation than seen in adult siblings (Fig. 6a)"; "The neurons from the two juvenile siblings exhibited considerable variation in spike waveforms (Fig. 6e, left and middle panels), and there was seemingly as much variation within birds as between the two birds (Fig. 6e, right panel)"; "First spike waveforms in cDAF birds also showed far more variation than expected in non-manipulated adult birds". None of these statements are backed by demonstration of statistical significance. It is unclear why this is so, maybe there was a lack of statistical power given the somewhat low number of recorded cells in each of the many comparisons performed? In some cases a statistical analysis was presented to exclude a confounding factor, such as in "There was no significant change in input resistance between neurons from non-manipulated animals and neurons from animals exposed to cDAF", but this still does not directly address the main issue here.

It is unclear why inhibitors of synaptic activity were not consistently used in all experiments, if the intent from the beginning was to isolate intrinsic properties of the recorded cells. Only rather late in the study do the authors present data on spiking activity under a cocktail of synaptic activity inhibitors. While they report no apparent changes in the 1st spike shape, it is unclear that there are no changes in spiking activity or in other spikes in a given current clamp recording. A more detailed assessment is needed here.

The attempt to directly demonstrate/characterize the proposed individual currents in the recorded cells was very limited. More specifically, a more concerted effort is missing to record under voltage clamp to isolate and biophysically characterize specific currents, and to use paired recordings before and after the application of pharmacological blockers, including statistical analysis, to more clearly show that specific currents are sensitive to individually applied drugs that can affect spiking behavior. This would be an important complement to the modelling effort, but it is missing in the study. If the modeling was based only on data from the previous literature, this should be stated more explicitly, and limitations in data interpretation acknowledged accordingly.

A related comment is that SK typically refers to a specific type of K^+ -sensitive Ca^{2+} channel. Why is only apamin (SK channel inhibitor) tested and not iberotoxin (BK channel inhibitor). Do the authors know that BK channels are not present in HVCx cells? This seems odd as the presence of BK channels has apparently not been tested, or at least it is not described in the literature for this song nucleus.

As acknowledged in the paper, the higher similarities in spike properties of cells from sibling pairs as opposed to random pairs of birds could be due to either shared genetic factors or similarities in upbringing/learning. An obvious test that was not performed would have been a cross fostering experiment where sibling pairs are raised by different fathers.

With regards to the juvenile data and predictions from HH models, some further analysis of the current clamp data that is missing in the study could provide some relevant tests for the model inferences as to possible current types involved. For example, one would expect differences in the sag of the input resistance and/or in the AP half width or repolarization rate when comparing juveniles vs

adults.

Given the acknowledged limitations in the dataset presented, the following is an overstatement that should be amended/toned down: "not via purely cell autonomous phenomena but in a manner sensitive to network and behavioral constraints."

In the DAF experiment, apparently no characterization of the songs from the experimentally manipulated birds was performed. Is the assumption here that it was not necessary to characterize the birds' songs because no changes to the songs were expected, given the prior literature? However, given the apparent changes that occurred in the intrinsic properties of HVCx cells, and the correlation of these properties with song bioacoustics that is being proposed by the authors, it seems like it would be critical to directly demonstrate that these birds did not change their songs. To take this point further, if no changes actually occurred in these songs of the DAF exposed birds, it seems hard to reconcile such a finding with the main conclusion that the parameters of HVCx cells reflect song representation, as proposed in the title of the paper. Furthermore, if the songs of the exposed birds did not change, then the statement that the observed changes in the intrinsic properties of HVCx cells are due to learning does not seem well supported.

Other specific comments:

Abstract:

...with non-synaptic mechanisms...: Not clear from the manuscript where this conclusion arose from, it seems highly speculative at this point. The authors should be more explicit when they are discussing a cell's intrinsic properties vs. changes in properties that presumably arise from altered synaptic communication.

'Ballistic singing' is used several times but not defined in the text.

The text states: "...all of which were clearly projecting towards Area X (Supplementary Fig. 1a), yet this figure does not depict axonal terminals in Area X, as the text seemed to imply.

'Plateau amplitude' seems to be defined only in Supplementary Figure 3, this should be made more clear in the main text and/or Figure 2 legend.

It is unclear why some other important spike parameters were not analyzed, such as AP half-width, or Max repolarization and depolarization rates. Such parameters might have been informative when analyzing cases where changes in K⁺-related currents are suspected, such as in juveniles or to test some predictions from the modelling effort.

Did input resistance vary across birds in correlation with the observed differences in spike properties? This seems relevant since input resistance can have significant effects in shaping active neuronal cell properties. A related question is whether the observed variability as stated in the text (181 +/- 35 MΩ) refers to SD or SE?

With regards to the statement: "Model fits were judged based on the response to current pulses of 100 pA...", what is the typical magnitude current from synaptic input onto HVCx neurons? Is 100 pA reasonable or appropriate?

With regards to the statement: "In our sample, model estimates of g_{Na} varied within the species by over 19.2x...", this seems like a very high level of variability, given that the range of variability in the

'first spike amplitudes' of actually recorded cells, as shown in Fig. 2c, is in the order of 35 mV or so. Is there any evidence that this would match variability in actual sodium currents recorded from real HVCx neurons?

The authors do not provide a definition or explanation of how they determined the 'threshold' values shown in Figure 3.

The recordings shown in Figure 4 provide evidence for spontaneous firing of HVCx neurons, however this aspect was not discussed or examined in any detail. Is spontaneous firing a consistent feature of these cells? Is this an aspect that is predicted by the model?

There are several concerns with regards to the data presented in Supplementary Figure 7.. As mentioned above pairwise recordings with individual pharmacological inhibitors and a more rigorous statistical approach could be used to test individual currents that the authors have determined to play a role in HVCx neuronal physiology. What was the intersweep holding potential for the voltage clamp experiments in (B-D)? (D) should include a dashed line or some indicator for 0pA. The blue sodium current isolated in (D) does not have the profile of a typical sodium current as there is a very small, slow inward current. The authors should consider addressing this current's profile or replacing it with a better example.

No description of the song bioacoustics analysis was presented in Methods.

Minor:

Abstract:

... within hours if not quicker... : no support for 'if not quicker', this should be deleted from statement.

Typo: ...HVC spiking activity do not change in zebra finches...

'Principal' component analysis, not 'principle'

The term IP is not defined in the text.

"N = 253 neurons arising from all adult birds with two or more HVCx neurons recorded"

Reviewers' comments:

We are grateful for the supportive comments and detailed reviews all three reviewers have provided. We are particularly grateful for the several comments that helped to clarify points of confusion on our part, and the many comments that helped to identify problems in the reporting. There are numerous edits to the manuscript and some changes to the figures based on the comments, and also some of the figures have been re-arranged as describe below. The introduction has a number of changes and the discussion is substantially re-written. Overall, the results and conclusions remain intact, while we believe the manuscript is greatly improved.

We provide both a clean version of the revised manuscript and one that shows all changes from the initial version.

Reviewer #1 (Remarks to the Author):

In their paper, Daou and Margoliash examine intrinsic properties of basal ganglia-projecting neurons in HVC of the zebra finch. Surprisingly, they find that responses to current steps administered to the cell body are considerably more similar across neurons within one bird than they are across birds. They use synaptic blockers to demonstrate that these responses are due to intrinsic properties of neurons rather than network effects. The homogeneity of intrinsic properties does not appear to exist in juvenile birds, and birds exposed to distorted auditory feedback show differences in their intrinsic properties compared to controls.

Overall, these findings are generally exciting, with the potential to add a major new insight to our understanding of the songbird brain as well as neural networks more broadly. However, the manuscript needs some work. Once the points below have been addressed in a satisfactory manner, I would be supportive of publication in Nature Communications.

MAJOR CONCERNS

(1) The weakest part of the paper is the attempt to model ionic conductances using a Hodgkin-Huxley approach. The attempt this using a single compartment model, despite the fact that an overwhelming amount of literature has found major intrinsic changes of ionic currents in the dendrites (and even single dendritic branches). It is unlikely that these changes will be easily detectable from somatic recordings, calling into question the utility of this modeling effort. For instance, the final paragraph of the discussion points to the importance of 'somatic intrinsic properties' (Page 19), which rings quite hollow in my view. Furthermore, parameters of the model are hand tuned and not verified using other approaches such as mRNA levels of specific channels (e.g., Shultz, Goillard, and Marder, Nat Neuro, 2006), immunostaining, or any other 'ground truth approach'. The reason that Eve Marder's work was compelling is that a large spectrum of models could explain the pyloric rhythm that was observed. Here only a single hand-tuned solution is presented, and that has the potential to be misleading. Therefore, as it stands, I think that the modeling component represents an unnecessary layer of obfuscation that should be removed wherever possible and replaced with primary observations, such as spike shape, spike rate, etc. For instance, I find Supplementary Figure 10 to be far more convincing than Figs. 6c&d – simply because the scientific point can be made with primary measurements alone rather than having to invoke speculative biophysical conductances.

We address the three separate issues raised here in the order raised.

We are aware of the literature finding intrinsic changes throughout dendrites (and the axon hillock as well) and extensively cited that literature in the introductory paragraph of the paper within limitations of space. We agree that additional measurements that we did not attempt would increase confidence localizing the site of the intrinsic changes, but that is not the focus or central to the present results. To alert the reader to this caveat we have added a sentence explicitly making the point (middle of second paragraph, p. 7). We also had reported that spike threshold was regulated within individual birds and varied among birds, which is a property that is more likely to arise from regulation of somatic (or axon hillock) currents. To help emphasize this point, which in the previous version of the manuscript was easy to overlook, we moved Fig. 2c (plateau amplitude v. first spike amplitude) to the supplemental information (now it is Supplementary Fig. 3), and moved Supplementary Fig. 3b (spike threshold v. first spike amplitude) to Fig. 2c. (We also fixed a Y-axis labeling problem in the spike threshold figure.) Furthermore, we also now clearly identify our speculations as such, at the end of the paper. More challenging, we are struggling with the reviewer's comments suggesting that the results we found likely arise from changes in dendrites but that these changes are unlikely to be detectable from somatic recordings. This seems to firmly shut the door on any discussion. We made somatic recordings. We saw variation. Through a number of experiments, we determined the variation was biologically meaningful. We modeled that variation. We then discovered that the variation aligned with features of the birds' songs. Several predictions that arise from that observation were subsequently confirmed.

It is an inaccurate and incomplete description of our results, one that promotes unwarranted confusion, to state that only a single hand-tuned solution is presented. The issue of hand tuning is mooted by the extensive study we conducted to directly address this issue. In that study we examined a vast number of possible solutions (exhaustive parameter search examining over two billion possible solutions per neuron using a supercomputer) for a large subset of the neurons. To our surprise there were consistent global minima identified exhaustively searching the parameter space, with the global minima varying from bird to bird. There were consistent results comparing hand tuning and automated parameter estimation, with simple explanations for when the two diverged. Hence, the modeling results do not reflect bias introduced by hand tuning. The fitted models also achieved very good to excellent predictions of the responses to other injected currents that had not been used in the fitting process. We believe that the reviewer should fully consider these results in evaluating the modeling and doing so should arrive at a different conclusion regarding the validity of the modeling. Since the hand tuning and the automatic parameter search found the same or highly similar global minima in the solution space, it is also inaccurate to assert that presenting the single solution was "misleading", an unfortunate choice of words. We focused on the best solution (global minimum) for each neuron, which surely is a reasonable result to report. Certainly it is possible that considering the large space of suboptimal solutions would yield additional insights (constraints) on network properties. Such an extensive modeling study may well be worthwhile. Efforts to estimate more HH parameters using data assimilation techniques arising from statistical physics (e.g. see work by Henry Abarbanel, that DM contributes to) are another approach. Both these represent separate modeling studies that are beyond the scope of this paper. We fully acknowledge, of course, that a five-conductance model of neurons recorded *in vitro* is an approximation of the dynamics the neuron expresses *in vivo*. Still, our modeling approach provides utility in analyzing the data of the paper and also, we believe that combining electrophysiology and such modeling is an uncommon and strong approach; it should be viewed as appealing.

The reviewer seems to imply that since our results, for a single solution describing the intrinsic properties of the population of HVCx in each bird, differ from Marder and colleagues' results, showing many solutions to explain the pyloric rhythm, this somehow brings to question our result. But that assumes that lobster STG pattern generation and songbird song system sensorimotor vocal learning operate under the same constraints. That is not an established fact. Indeed, our results argue to the contrary. Our work relates intrinsic properties to individual-specific highly skilled learned sensorimotor behavior of animals. Many aspects of the singing behavior in songbirds and digestive STG rhythms differ (e.g. rate, complexity, timing, precision, known degree of learning). Given that there are different biological constraints driving organization of the STG and the HVC, surely it is a worthy result to have characterized some of the physiological differences that arise. To put it another way, here too “a large spectrum of models [can] explain the [HVC] rhythm[s] that [were] observed” but in songbirds each model is associated with one bird. Is that not an interesting and plausible effect of complex sensorimotor vocal learning, where each bird has its own unique song?

Most importantly, however, we respectfully strongly disagree that the ionic conductance models are weak or represent “obfuscation”, another unfortunate choice of words. We agree that the raw data provide useful information, hence we provide extensive analyses of data measured directly from the recordings, both in the figures and supplementary information. We demonstrate the principal results with information extracted from raw data for spike trains and spike waveforms (Figs. 1 and 2, Supplementary Fig 3) and we demonstrate the similarity of results for siblings also with information extracted from raw data (Figure 6 and Supplementary Fig. 9). We also show model fits with exemplar raw data from four neurons (Fig. 3), exemplar raw data showing the effects of isolating the neurons from the network (Fig. 4), multiple spike trains comparing two adult siblings (one exemplar pair of four adult pairs) and two juvenile siblings (two exemplar juveniles of four juveniles) (Fig. 6), and raw data spike trains from an exemplar DAF bird (Fig. 7). Our paper lacks neither from analysis of raw data nor extensive exemplar raw data, all of which support our conclusions.

But quite to the contrary of obfuscation, the modeling provides a biologically plausible mechanistic explanation for what is changing in the HVCx – that is, what changes in the magnitude of a set of intrinsic currents that have previously been pharmacologically demonstrated for HVCx could give rise to the changes in spike waveforms and spike trains that were observed. The raw data do not provide that mechanistic insight. Beyond that, it is important to note that estimating the conductances proved central to finding perhaps the most compelling result reported - a direct relation between the cellular properties and singing behavior of the animals, lending support to the utility of this approach. It is easy to defend our choices of ion currents that we modeled that give rise to that result - those currents have been shown to exist in HVCx. Doing these analyses for an arbitrarily chosen set of parameters from the raw data is less attractive. Also, potentially having to search for a subset of raw data attributes that yield a particular result may give concerns regarding arbitrary choice of parameters and statistical validity. Moreover, the insights from the mathematical modeling was the main factor that motivated the voltage clamp experiments (Supplementary Fig. 8), which provided additional and independent support of the central results. The modeling also provides a concise way to describe the variation of HVCx in individual animals, which was helpful for comparing juveniles and adults, and for comparing DAF adults from normal adults. The modeling also allows us to factor out network effects that may arise through the gSK calculations. To be sure, our modeling approach – any modeling approach – also has limitations and simplifications, which here we

have attempted to respect in our reporting. In conclusion, given the utility of the modeling approach that we demonstrate, we respectfully decline to remove or diminish it from the paper. Quite the contrary, we believe it is a manifest strength not a weakness of the paper. In trying to address the reviewer's concern, however, we now alert the reader at several points in the paper to Daou et al. 2013 which pharmacologically confirmed HVCx express the five ion currents we are modeling, and where the relation between HVCx conductance magnitudes and spike waveform features was extensively described.

Finally, we are in full agreement that scRNA, or viral techniques to manipulate specific channels, or more sophisticated single cell electrophysiology are exceedingly exciting follow-up studies to pursue given our results - as are many others that are now open for pursuit. It is true there are many important and unanswered questions that our study has raised. Is that not the mark of good science?

(2) In the methods, it is stated that 'Most recordings were at room temperature' (Page 20). The authors should clarify this statement. For instance, if different birds were recorded at different temperatures, then this could easily explain the primary phenomenon described.

Thank you for the comment, we had not been attentive to this detail in our initial reporting and it strengthens the paper to elaborate on the effects of temperature. We find no correlation between temperature and the clustering of spiking properties. The great majority of the data were collected between 24.5 °C – 25.5 °C, and these recordings show extensive variation in estimated ion current magnitudes (or raw data features). There is, of course, an effect of temperature on the physiological properties we recorded. For example, for two of the five modeled

conductances (I_K and I_h), the correlation between conductance and temperature is significant, but for all five conductances the relationships are noisy (see five plots, above). The temperature does not explain the clustering of physiological properties on a per-bird basis, as is clear from inspection of the graphs above. For example, if we compare all pairs of birds, plotting the difference in conductance space centroid positions as a function of the absolute temperature

difference (using absolute temperature since distance is always positive), there is no relation (see plot, below).

These data derive from all 68 neurons (16 birds) for which we have exact temperature data. Unfortunately, a technical error (failure to include code to record temperature when the protocols were ungraded in the course of these experiments) precludes us from reporting for more birds. However, we have no reason to believe there were systematic differences in the temperature the other experiments were conducted at. Also, these 68 neurons are distributed across almost the entirety of the range of estimated ion current magnitudes for the full data set. Thus, we fail to find a temperature relationship that "easily explains the primary phenomenon described" in neurons that span the full range of variation expressed in the full data set.

We have modified the methods accordingly to report these results (paragraph starting at the bottom of p. 21).

(3) At several points in the paper, the degree of speculation outpaces the amount of available data. For instance, the 'long-sought physiological signal' is misleading. Although the changes in HVC(X) neurons are intriguing, we don't yet know the impact of these changes. For instance, it is difficult to think of a method to selectively halt the changes in intrinsic properties in those neurons to determine the effect on the network and behavior. Until that time, we are left with a correlative result whose significance (or lack thereof) will be determined at a future point. This does not take anything away from the present finding, but too much speculation may serve to distract from what is otherwise a fascinating empirical observation.

We wrote "a" not "the" long-sought physiological signal, "that rapidly changes with altered auditory feedback". We believe that our statement is factually correct, not misleading. Yet it is not our claim that we have a mechanistic understanding how HVCx IP are altered by singing behavior – that we have found the signal, the explanation. We do believe we have made some inroads, e.g. see comments of other reviewers. We have altered the language attempting to avoid any such confusion (p. 2). We agree that too much speculation can be distracting and

have revised the entire Discussion accordingly. Please also see below (point 5). Finally, yes this is correlational, but the correlation is strong: finding the relation among normal adult birds led us to predict the positive results subsequently obtained with DAF birds, adult siblings, and the developmental study (which was conducted prior to Ross et al. 2017). To go beyond that, we are fully in concordance that we need to directly manipulate HVCx intrinsic properties, and we are working on approaches to do that.

(4) Very recently, a paper was published from Richard Hyson's group showing the impact of tutor song experience on intrinsic properties of HVC neurons (Ross et al., J Neurosci, 2018⁹). Those data are broadly consistent with the work presented here without stealing away any of the major thunder. That manuscript should be cited and thoroughly discussed. Also, given that paper and Hyson's previous work, the authors should say that their data 'confirm' (rather than 'indicate') that intrinsic properties are changing through development (Page 14).

We have added the following paragraph to the thoroughly revised Discussion:

HVC_x IP are also regulated over longer time scales. A recent study^{Ross 17} demonstrates that HVC_x sag and rebound properties show developmental changes and variability during development. We confirmed those results and extended them to include other features of IP, spike waveforms and excitability (numbers of emitted spikes). In a subsequent study it was reported that untutored juvenile birds exhibited properties similar to adults, but that tutored juvenile birds differed from both, a seemingly paradoxical result^{Ross 19}. We too observed differences between tutored juvenile birds and adults (Fig. 6f), and our results may help resolve the reported paradox. Given that the prior studies focused on recording from multiple classes of HVC neurons, only few cells were recorded per cell class. This precluded sampling sufficient HVC_x to detect the homogeneity of intrinsic properties within individual adult birds as reported here and left unresolved the distribution of HVC_x IP in individual untutored juvenile birds. If prior to tutoring, HVC_x intrinsic properties are broadly dispersed with each juvenile, this would not have been detected given the sample sizes of 1-3 HVC_x /bird^{Ross 19}, and small samples of neurons per bird in untutored and adult birds would both appear broadly and similarly dispersed, resolving the paradox. Future studies exploring variability of intrinsic properties within untutored and tutored juvenile birds may be a compelling approach to elucidating genetic and epigenetic features of developmental song learning.

(5) Overall, the manuscript would benefit from careful proofreading in order to clarify the progression and interpretation of experiments and to improve logical flow. For instance, on Page 2, the following fact are presented: (i) Zebra finches do not modify the basic song motif structure rapidly in response to perturbations, (ii) Zebra finches require auditory feedback to maintain their songs, (iii) Therefore, we need to find the parameter that is regulated. This is not fair. Of course, auditory feedback does not lead to major song changes at the level of motif structure, but conditional auditory feedback (Tumer et al., 2007) can lead to lawful pitch (Andalman et al., 2009) and timing (Ali et al., 2013) shifts in zebra finches that are likely due to spiking changes in LMAN and HVC, respectively. There is no need to create a vacuum that intrinsic property changes must fill.

As we mention in the introduction, unlike zebra finches Bengalese finch show immediate and rapid changes in behavior and in HVC activity in response to auditory feedback challenges, so the report by Tumer and Brainard 2007 using Bengalese finches is not germane. It does not directly illuminate whatever differences in physiological mechanisms that might obtain in zebra finches. Andalman et al. 2009 showed adaptive pitch changes in zebra finches, but over the time course of several hours at best, and in any case their focus was on pitch not responses to

changes the timing of feedback which we explore. Ali et al. 2013 showed adaptive timing changes in singing behavior that evolved over days.

Importantly, here the discussion is to find physiological changes that correlate with the response to the altered auditory feedback. The reviewer asserts that shifts in singing behavior “are likely due to changes in ... HVC”. We agree – and importantly, what changes in physiology? All the prior studies in adult zebra finches that specifically searched for such immediate physiological signals in single neurons after manipulating auditory feedback failed to find them (e.g. HVC: Kozhevnikov and Fee 2007, Hamaguchi et al. 2014, Vallentin and Long 2015; IMAN: Leonardo 2004). Margoliash (1986 JNsci) reported auditory responses in awake white crowned sparrows. Dave and Margoliash (1998 Science) reported auditory responses in sleeping but not awake zebra finches. Prather (2013 Hear Res), examining HVC auditory activity in awake birds reported positive results for swamp sparrows, Bengalese finches, and canaries (there are other species as well), but could not find them in zebra finches. He concluded “This stark difference suggests that different species may have solved the challenge of sensorimotor comparison in different ways.” Similar or related comments are found in the other papers we cite above, and throughout the literature.

Thus, there is a disconnect – in zebra finches – between the known changes in behavior under conditions of perturbed auditory feedback and the lack of knowledge how single neurons change under such conditions. Hence, we disagree that there is no gap in knowledge understanding how the moment-to-moment changes in auditory feedback are encoded and retained in zebra finches. Of course, we are not in a position to assert that intrinsic properties have “filled” that vacuum, nor did we make such speculation. Nevertheless, a reasonable and important speculation that arises from our results is that changes in intrinsic properties are part of that mechanism. Other reviewers support the same point. We take the reviewer’s comments on this point as our failure in reporting and have carefully modified the ms to clarify this issue.

(6) On Page 3, it is stated that 76% of neurons had a RMP $\leq -70\text{mV}$. Rather than provide this number, I would rather see the mean and variance of the RMP for all recorded neurons as well as the range, mean and variance of the current needed to hold neurons at -70mV .

We now report those data as well (top of p. 3), RMP at break-in $-70.33 \pm 1.77 \text{ mV}$ (mean \pm SD), holding current $9.73 \pm 16.35 \text{ pA}$, min -30 pA ; max 35 pA .

(7) In Figure 32, the 'plateau amplitudes' for individual neurons are quantified, but the associated raw data are not shown. It would be considerably easier to understand the figure if these traces were included for example neurons.

We have now added an inset to show examples from three neurons, each from a different bird. We also note in the figure legend that the plateau amplitude is explicitly defined in Supplementary Figure 2. Please also note that we have moved the plateau potential distribution to Supplementary Fig 3 and moved the spike threshold distribution to Fig. 2c.

(8) On Page 5, it was stated that input resistance varied from neuron to neuron 'as expected'. I find this absolutely *unexpected* given the results presented for plateau amplitude, spike rate, and interspike interval – all factors that should absolutely depend on input resistance. The authors should provide a clear explanation for this.

This was an exceedingly useful comment that cleared up a confusion we have labored under for some time. We had made a preliminary statistical analysis at one point correctly calculating means and variance but incorrectly analyzing variation across animals and accepted this result without more careful consideration of the conclusion that input resistance varies in a noise-like fashion. Yet it was nagging. We are most deeply grateful to the reviewer for pointing out that this conclusion was incorrect, motivating the analysis that follows.

Revisiting this issue, indeed we find correlations between input resistance and features of the response, for example we examined first spike amplitude. We see significant correlations examining all neurons in all normal (non-DAF) birds and when examining the average values for each bird (meaningful since our hypothesis is that the neurons cluster). In the 8 DAF birds, where the clustering of physiological properties is disrupted there might be residual correlation, but it does not achieve significance (see graphs below). We have modified the text to reflect these facts. We are most grateful to the reviewer for the insight.

(9) The authors state that they administered hyperpolarizing current traces in addition to their 'canonical' 100pA depolarizing current traces. These hyperpolarizing current traces and associated data (e.g., membrane potential 'sag') should be used to further characterize the intrinsic properties of neurons presented here.

In all phases of the experiments we used both the hyperpolarizing and depolarizing pulses to characterize the intrinsic properties. We have clarified this point in the text.

(10) To help address the overall concern about other uncontrolled sources of experimental variance, the authors should examine the relationship between intrinsic properties and the series resistance measurement for each recording. If recording quality has nothing to do with the results obtained, then there should be no relationship.

There is no relationship. See plot below. We have added text to indicate there was no relation between series resistance and first spike amplitude.

(11) On Page 10, the authors engage in some analytical acrobatics that leave me a bit concerned. There, they mention that 'one outlier bird gave rise to all the non-significant results', and once those data were removed 'all but one were significant'. I don't see the benefit of curating the data in this way, and the authors should remove this from their analysis.

We respectfully disagree. We treat this feature of the observed data in standard fashion, reporting the correlation and statistical significance with and without the outlier. There is nothing acrobatic about this analysis, rather it is well-grounded. We consider both the entire population of songs and neuronal conductances at once, and each bird one at a time compared to all the other birds. Considering the entire population, the results (song similarity vs. conductance distance) are highly significant with or without the outlier removed. Considering each bird against all the others, 11/18 are significant, but this rises to 15/17 (with the 16th bird almost significant) after removing the outlier. That is, one bird is giving rise to much more noise in the data set than is any other. We provide a plausible explanation, that the outlier is one of only two birds with only three cells recorded, which will decrease the confidence of the mean values obtained for those two birds. In our opinion this information is useful for the reader to evaluate the biological (not statistical) significance of the results considering individual animals, not just populations means. And given how central to the paper is this result relating cellular properties with individual, learned singing behavior, it is in our opinion worth elucidating. Of course, the reader is free to ignore the results for individual pairwise comparisons with the outlier removed, or not.

In summary, we believe not including the short sentence "Furthermore, one outlier bird gave rise to all the non-significant results" would represent a failure of reporting. This is followed by only a single sentence to explicate this conclusion. We hope that the reviewer reconsiders.

(12) The estimation of the number of motifs needed during DAF is somewhat flawed. As the authors point out, this relationship is not necessarily logarithmic and appears to be strongly influenced by a single point (Figure 7d). More data are needed to better understand this relationship, and – until that happens – the claims should be softened accordingly.

It is incorrect that the fit is highly influenced by a single point. See Fig. 7d inset (lin-log plot) – which is why we included that inset. The straight-line fit passes close to the 4-hour point. That is, the 4-hour point is not an outlier. (Two other points lie further from the line than the 4-hour point.) We have now expanded our description of the Fig. 7d inset to better help the reader to devaluate the fit.

Removing the 4-hr point (for which there is no justification) would only slightly modify (reduce) the estimated number of motifs before a bird started to change IP in response to DAF. It is easy for us to calculate the estimate with the 4-hour point removed, but do not see how we could justify reporting this result.

(13) Also, the authors do not quantify how much degradation results from the DAF – how was this related to changes in intrinsic properties?

We have added the observation demonstrating that the zebra finches exposed to cDAF decrease the amount of singing and increase the number of vocalizations (calls and introductory notes) just before the onset of song. There is a much more complex story here to describe the behavioral changes, which we are preparing for publication, being developed by a graduate student in DM's laboratory.

We do not have a measure of the behavior which covaries with IP except in the following sense. Note that there is no solution available to the birds to cancel the abnormal feedback except to stay mute. (Indeed, the amount of singing is suppressed following the onset of cDAF.) We believe that this helps induce the rapid changes that we observed and will continue to provide an error signal whenever the bird sings. Hence error should accumulate over time, as we have observed.

MINOR CONCERNS

(1) The results relating to run-down should be transferred from the Methods to the Results, specifically the lack of impact of the number of days between experiments or the variation in measured responses throughout the duration of the experiment.

We state the results in the text (p. 7) and provide a clear pointer to the analysis for the interested reader. Also, there was no time-dependent variation in the IP values over the duration of an experiment, or the interval (days) between experiments (see Methods: Run-down).

The Run-down section is lengthy. Concurring with this request would devote considerable space of the main text to addressing a control analysis. Alternatively, we would pare away the language and ultimately provide an overly condensed, hard-to-understand description. Given these considerations we believe the reader is best served leaving the description of run-down as supplementary information and respectfully decline the suggestion.

(2) 'IP' is used throughout the text without being defined at any point.

Thank you, fixed.

(3) On Page 10 (third line from bottom), the value for R is missing.

Thank you, fixed.

(4) On Page 17, the authors speculate that populations of HVC(X) neurons could be related to different syllable types. Given the homogeneity of intrinsic properties, how would these populations be defined?

Within the homogeneity of intrinsic properties observed for a given bird, there could also be more subtle inhomogeneity (e.g. based on syllable) that could only be detected using a different experimental design. That is, we recorded ≤ 10 HVCx per animal, whose song might include 3-8 syllables per motif. It would be difficult to see per syllable variation under these constraints of the experimental design. Unfortunately, however, this comment was dropped from the discussion to provide for better focus on the major points.

(5) On Page 17, the authors speculate that 'common total synaptic drive' could induce common values for 'intrinsic properties'. If this is true, then that suggests that HVC neurons whose intrinsic properties are heterogeneous (e.g., juveniles or adults following DAF) have heterogeneous synaptic input? That is a strong – and testable – prediction.

Agreed. But this too was dropped from the discussion to better focus on the major points.

(6) On Page 17, the statement that 'HVC(X) spiking is characterized by hyperpolarization during singing' is confusing: How can spiking be characterized by hyperpolarization?

Wording is fixed; HVCx hyperpolarize prior to the onset of singing.

(7) Debanne has a wonderful new review on plasticity in intrinsic properties (Debanne et al., 2019) that should be cited here.

We have done so, in the concluding discussion paragraph. This review includes a discussion of how regulation of spike threshold leads to a “global” change in a cell’s output, as we are proposing.

(8) The statement that HVC(X) and HVC(RA) neurons project exclusively to X and RA, respectively is not formally true. In fact, in a recent paper from Mike Long's lab, 2 out of 40 projection fully reconstructed HVC projection neurons sent axons to both structures (Benezra et al., JCN, 2018).

Correct. This is now noted in manuscript (Fig. 1). For convenience we define HVC_x and HVC_{RA} as those neurons that project exclusively to Area X or RA, and don't give a name to those projecting to both targets.

Reviewer #2 (Remarks to the Author):

This is an exciting study showing that intrinsic excitability properties in a subtype of neuron in the avian song circuit critical for vocal learning are intricately aligned with the type of song that the bird produces. By measuring a set of 5 ionic currents in these neurons and using modeling to confirm that these currents are sufficient to predict underlying cellular excitability, the authors show convincingly that the ionic conductances that determine a neuron's specific intrinsic excitability are much more similar between neurons in an individual than they are across individuals and that the similarity between individuals is highly correlated with song relatedness.

These findings suggest that intrinsic excitability is tightly linked with the neural, and by extension circuit, demands for the songs that individual birds produce.

Thank you for these comments. We concur that there is utility for estimating the ion conductances and using these in modeling the response of the neurons.

While this finding is surprising and highly interesting in its own right, one of the most remarkable aspects of the study is the finding that excitability profiles are highly plastic and that sensory perturbation for as little as 4 hours (using continuous delayed auditory feedback) will cause a change in these highly stable excitability profiles. Because the neurons where this change is observed (HVC neurons that project to the basal ganglia) play a critical role in vocal plasticity, these findings imply that intrinsic excitability might be one of the first targets for inducing vocal plasticity following sensory feedback perturbation. The observation that juvenile birds during the song learning stage lack the high degree of homogeneity in intrinsic excitability that is observed in adults, further suggests that the regulation of intrinsic ionic properties is fundamental to the vocal learning process. These are significant findings because they offer a new way of thinking about circuit-level plasticity which tends to be highly focused on changes in synaptic weights. While there is a small but growing literature showing that changes in the expression pattern of specific ionic conductances can be intimately linked to neural plasticity, this study is the first to show the importance of intrinsic excitability and modulation within the context of a complex learned natural behavior.

Overall I found this to be a thorough and well-crafted study that combines experimental and computational approaches. The results section of the paper is clear and easy to follow. My biggest issue with the current manuscript is the discussion section which is difficult to read and generally does not do this study justice. The figures are generally good but could benefit from some minor modifications to make certain points clearer. Although not necessary, in my opinion, for this first description of the phenomenon, it would have been nice if the authors could have used genetic relatedness to further highlight the point that similarity in intrinsic excitability is related to the type of song the bird produces rather than heritability of the trait. In future experiments it would be interesting to compare intrinsic excitability in siblings tutored with different songs. It would also be interesting to compare excitability profiles in adult siblings singing highly similar songs (as in figure 6c) but with one sibling experiencing cDAf while the other is treated to a neutral control condition.

The suggested experiments are ones that we are currently pursuing. We concur that the discussion was confusing. The discussion suffered from trying to say too much, covering too much ground, that is better left for a review. We have largely re-written the discussion, simplifying it and hopefully clearly reporting on a smaller number of central results.

General Comments:

Discussion:

The discussion section would benefit significantly from an overall re-organization. As written it seems to lose the thread that is so clearly laid out in the results section. The discussion also contains a number of long sentences that feel grammatically awkward and therefore difficult to follow. In general, the discussion seems to jump around quite a bit and is quite heavy on "song system" jargon which will make it difficult for a reader to follow if he/she does not have the

appropriate background. The lack of a figure of the neural circuit makes it particularly difficult to follow some of the more complex arguments that the authors lay out in terms of mechanisms that might drive the observed similarity in intrinsic excitability.

We moved the song system figure from Supplementary to Fig 1 and improved upon it. Per comments above, the discussion section is re-written.

Figures:

Aside from adding a "song system" figure that could be used to illustrate some of the arguments made in the discussion section, I am wondering if the authors could come up with a more effective graphical representation to show the changes that occur when young juveniles transition to adulthood. The same suggestion holds for showing the changes that occur following cDAF. Figure 6c&F as well as

We tried several approaches to summarize the principal results and were not satisfied that they helped improve the reporting; ultimately, we abandoned this effort.

figure 7c are particularly difficult to visualize especially on the low quality pdf because the low intensity symbols are hardly visible and even if they were, the full meaning of the finding lacks that immediate pop out effect.

We agree. Even with a high res pdf still it was difficult to achieve a proper "background" level of visibility for the symbols. Instead, we have modified Fig. 3c to surround the distribution of points for all normal birds/cells with a dotted line, then imported that line onto the data in Fig. 6c/f and Fig. 7c. This allows us to remove the background symbols while improving the ability of the reader to directly and immediately compare the distributions in Fig. 6c/f and 7c with the distribution from Fig. 3c. This reduced clutter and improves the readability of the figures.

Minor comments:

Abstract: I recommend removing the word "if not quicker" in the statement "degraded, within hours if not quicker, in adults singing" given that the authors do not provide any evidence for such claim.

We modified the Abstract.

Page 2 – bottom

For the following statement "Furthermore, in response to feedback perturbation during singing HVC spiking activity do not change in zebra finches whereas reliably changes are observed", the authors should also include several references showing that such perturbations do not affect synaptic activity (e.g. Hamaguchi et al. 2014 and Vallentin and Long, 2015). There is also a typo in that sentence.

References added and typo fixed.

Page 3 – top

It would be helpful to add a short half sentence here (details can remain in the supplemental section) stating that neurons are retrogradely labeled with tracer following tracer injection in Area X. This might also be a great place a modified figure of the song system. This figure can then serve the dual purpose of highlighting the methods as well as serving as a reference for the various arguments in the discussion.

We believe the reviewer may have missed the short sentence reporting the 114 cells that were retrogradely labeled following Area X tracer injections. We have moved the song system figure to Fig 1 and refer to it here, which indeed will help the reader understand the design of the experiments.

Figure 2

In figure 2b it would be helpful to indicate that each color represents the average spike waveform for different birds. **Done.** Would it be possible in this figure to also include some measure of spike waveform variance by including +/- SD contours? **We now show the variance at peak amplitude and spike threshold. We tried showing SD contours over the entire waveforms, but this swamped the traces leaving only the top-most trace visible.**

Page 6 – middle

The following definition for the SK current as "small potassium-mediated calcium channels (ISK)" is awkward. It almost reads as if it were a calcium current. Why not describe it simply as a calcium-dependent potassium channel?

Done.

Page 8 – top and Figure 3b

This figure is difficult to interpret. The rationale for this figure should be revised in the text and the figure itself should be modified. In particular, it is quite difficult to notice the small arrow and circle on the bottom right of the figure. It is also not clear how to interpret the error axis. What is the significance of an error value of 400? More detail needs to be provided in the legend as well as the text.

Done. Most of the additional explanation is in the text. The error values are arbitrary units, now identified as such.

Figure 6c

In addition to the difficulty of seeing the shaded symbols (see comment above), the definition of the different birds was a bit confusing to me. What threw me was the sibling pair with different (leg-band I assume) names (Gray15 and White3). Why not simply relabel symbols as (sibling 1 & sibling 2; parent colorX)

(per above) the shaded symbols have been replaced by a dotted line delineating the boundary defined by the data in Fig 3c. The sibling (and juvenile) bird in Fig 6 have been renamed along the lines suggested, which improves readability.

Figure 7

The figure legend is too sparse to provide a clear explanation. This is especially true for figure 7c. Figure 7d is also difficult to interpret because the legend indicates that the histogram represents changes on the first day of cDAF exposure, yet several of the data points are from birds that are described in figure 7b to have been exposed to cDAF for two days or more.

Fixed.

Page 16 – bottom

A few examples of confusing and/or grammatically awkward sentences that are rather typical for the discussion section

"Many songbird species exhibit within-individual variability in their adult singing behavior, not only well known cases such as Bengalese finches, canaries, mockingbirds, and starlings, but including at least some species that are commonly thought of as having a single song repertoire (e.g., indigo buntings), and also in response to seasonal changes and other factors."

"In zebra finches, song output and song system neurons suprathreshold activity was found to be refractory to acutely altered auditory feedback as well as manipulations of other forms of sensory feedback, albeit cDAF was not tested."

These sentences were dropped in the general re-write of the Discussion.

Reviewer #3 (Remarks to the Author):

This is an ambitious study that addresses important but still largely unexplored questions in birdsong biology. The authors have conducted detailed recordings in slices and modeling efforts to characterize the intrinsic physiological properties of HVCx neurons, a key component of the song control circuitry representing the source of input from the vocal motor pathway into the anterior pathway for vocal learning. They provide substantial evidence that these cells have similar spike waveform properties and spike train timing within individual birds, but also that these properties vary considerably across individuals, apparently in correlation with individual variability in song bioacoustics. The study also provides evidence of higher variability of these parameters in juveniles at an age when their songs are not yet highly stereotyped, and that these parameters seem to change in adults as a result of auditory feedback disruption. These findings are presented as evidence for song and error representation in zebra finches. Overall, the effort described is quite substantial and the study is a potentially important contribution, however some significant weaknesses decrease enthusiasm for the paper as presented, both in terms of data and interpretation. The main issues are listed below, followed by more specific comments and minor points.

Several statements and conclusions are not backed by rigorous statistical analysis and appear more like qualitative assessments of the data. This includes statements like [1] "spike amplitude and plateau amplitude clearly segregated neurons from different birds"; [2] "the model of a given neuron was far better at predicting responses of that neuron than were models of neurons from other birds"; [3] "there was a substantial variation in the trains of spikes emitted by different cells

within each of the two juvenile siblings (Fig. 6d), far greater variation than seen in adult siblings (Fig. 6a)"; [4] "The neurons from the two juvenile siblings exhibited considerable variation in spike waveforms (Fig. 6e, left and middle panels), and there was seemingly as much variation within birds as between the two birds (Fig. 6e, right panel)"; [5] "First spike waveforms in cDAF birds also showed far more variation than expected in non-manipulated adult birds". None of these statements are backed by demonstration of statistical significance. It is unclear why this is so, maybe there was a lack of statistical power given the somewhat low number of recorded cells in each the many comparisons performed? In some cases a statistical analysis was presented to exclude a confounding factor, such as in [6] "There was no significant change in input resistance between neurons from non-manipulated animals and neurons from animals exposed to cDAF", but this still does not directly address the main issue here.

These statements were indeed meant to be qualitative assessments of the data, to help facilitate visual inspection of striking features of the distribution the data. We now report supporting statistical analysis, as follows. To maintain the flow of the text, we have collected several of these analyses ([1], [3], [4], and [5]) in a new Methods section, Additional statistical analyses.

[1] Fig. 2c. Clustering in the spike threshold/spike amplitude space. For each neuron in each bird with $N > 2$ neurons, we calculated the distance (in the 2D space of spike amplitudes and spike thresholds) to another randomly chosen neuron for that same bird and for a randomly chosen neuron in another bird. Considering all 241 neurons, the same-bird distances were smaller than the across-bird distances, and the result was highly significant ($t = -18.9$, $df = 240$, $p = 1.5 \times 10^{-49}$). This confirms the visual impression that neurons from the same bird were systematically closer to each other (clustered) in the space compared to neurons from other birds.

Supplemental Fig. 3. Clustering in the plateau amplitude/spike amplitude space. The equivalent analysis as for Fig. 2c (above). Considering all 241 neurons, the same-bird distances were smaller than the across-bird distances, and the result was significant ($t = -18.8$, $df = 240$, $p = 3.5 \times 10^{-49}$), confirming the visual impression as per logic above.

[2] That "the model of a given neuron was far better at predicting responses of that neuron than were models of neurons from other birds" was already compellingly supported by extensive cross-validation statistical analysis reported in Supplemental information. We now alert the reader to that. We have added the following graphic (Supplemental Fig. 5) providing visual support to this claim.

[3] *Fig. 6d.* Comparing spike trains in sibling juvenile and sibling adult birds. For each sibling bird, we separated the cells into groups by the number of spikes emitted. Any group with only one cell was discarded. For all cells in each remaining group, we computed all pairwise ratios of times (relative to the start of the stimulus) comparing first spikes to each other, second spikes to each other, etc. and accumulated these values across all groups for each bird. We accumulated the distribution of spike time ratios (first spike i / first spike j , second spike i / second spike j , etc.) across each of the two juvenile siblings then compared each juvenile against each of six adult siblings (eliminating the two adult siblings with only two cells per bird). All 12 comparisons were highly significantly different, with the two-sample F test statistic ranging from $F(29,23) = 59.3$ to $F(20,23) = 3.3 \times 10^3$, and $P = 2.0 \times 10^{-15}$ or smaller. This demonstrates that the timing of spike trains in juvenile birds was consistently more variable than the timing of spike trains in adult birds.

Fig. 6e. Comparing spike waveforms in sibling juvenile and sibling adult birds. For the two juvenile siblings (6 cells recorded in each bird), we found the mean squared difference between one cell's first spike and the first spikes of all cells in the other bird, yielding 36 values. We then did the equivalent analysis for all four pairs of adult siblings. Comparing the juvenile pair with each of the adult pair, the two-sample F test statistic ranged from $F(29,35) = 0.0017$ to $F(9,35) = 3.6 \times 10^{-4}$, with the largest $P < 8 \times 10^{-12}$. In contrast 3 of 6 adult siblings were not significantly different from each other, and for the other three comparisons $P = 0.0015$ was the smallest value. This supports the visual impression that the variation among juveniles was much larger than among adults.

[5] *cDAF.* Spike waveforms in cDAF birds have much more variation than in non-manipulated adults. Within each of the eight normal adult birds with ≥ 8 cells per bird, we computed mean squared spike waveform differences between all pairs of first spikes. These were all pooled together as the variation expected for non-manipulated birds. We then did the equivalent analysis on each of the seven adult DAF birds. Comparing the pooled normal adult data against each of the DAF birds using a two-sample F test yielded $P < 2 \times 10^{-59}$ or smaller. Comparing the pooled normal adult and pooled adult DAF data yielded $F(242,138) = 0.0022$, $P = 2.9 \times 10^{-245}$.

[6] Regarding analysis of input resistance, please see the revised analysis (major point [8] Rev 1). We mention the lack of change in input resistance for cDAF birds as a point of reference to compare with normal birds.

It is unclear why inhibitors of synaptic activity were not consistently used in all experiments, if the intent from the beginning was to isolate intrinsic properties of the recorded cells. Only rather late in the study do the authors present data on spiking activity under a cocktail of synaptic activity inhibitors. While they report no apparent changes in the 1st spike shape, it is unclear that there are no changes in spiking activity or in other spikes in a given current clamp recording. A more detailed assessment is needed here.

We recorded only one cell in a slice that was bathed with synaptic blockers for fear that washout was incomplete (i.e. recordings of subsequent cells could potentially be affected by residual drugs or changes in the health of the slice). This point is now made explicitly in the ms. If for example, we had recorded only one cell per slice our sample size would be roughly 1/5 as large.

Using that approach we would have only recorded on average two or maybe three HVCx per bird, perhaps totally obscuring or in any case dramatically decreasing the confidence of the principal result, which is most compelling in the birds with many cells recorded per animal. We note elsewhere (major point [4], Rev 1) that limited sample sizes per cell type per animal characterizes the Ross et al. 2017, 2019 papers and this constrained the conclusions they reported.

Furthermore, while first spikes shape remained unaltered, we clearly reported that there are changes in response including changes in the timing of spikes in bursts following application of inhibitors of synaptic activity (Fig. 4). We explain these as changes in our estimate of gSK, which is a proxy for inhibition that arises from the activity of inhibitory neurons when synaptic activity is not blocked. In the charge balance equation that we used, loss of inhibition will be modeled as decrease in gSK, i.e. increased excitation of the cell.

The thrust of the reviewer's comment is noted, however. In principle our assessment of song similarity vs. distance in conductance space is contaminated by a contribution of network activity hiding in the gSK estimates. This appears to be a small contribution that does not show much variation from animal to animal (see Fig. 4), and since all the currents covaried across birds this should not dominate the reported correlation. Nevertheless, to directly respond to the comment of the reviewer, we recalculated the correlation between song similarity and distance in conductance space after having removed gSK from the conductance distance calculation. The correlation remained high both for the Euclidean and for the Mahalanobis distance calculations, and the results remained highly significant, also demonstrating that whatever network effect is creeping into our calculation via the gSK estimates this does not dominate the resulting correlations. These results are now reported in the text (bottom of p. 11).

The attempt to directly demonstrate/characterize the proposed individual currents in the recorded cells was very limited. More specifically, a more concerted effort is missing to record under voltage clamp to isolate and biophysically characterize specific currents, and to use paired recordings before and after the application of pharmacological blockers, including statistical analysis, to more clearly show that specific currents are sensitive to individually applied drugs that can affect spiking behavior. This would be an important complement to the modelling effort, but it is missing in the study. If the modeling was based only on data from the previous literature, this should be stated more explicitly, and limitations in data interpretation acknowledged accordingly.

Respectfully, we believe that this may represent a misunderstanding on the reviewer's part. As we stated in the paper, the ion currents that we modeled have been rigorously analyzed in prior work, see especially Daou et al. 2013 JNeurophys. There, among other results, authors used pharmacological approaches to demonstrate that each of the five ion current species are represented in HVC_x neurons, and also developed the Hodgkin-Huxley modeling that is the basis for our analyses here. The single compartment HH model developed in Daou et al. 2013 did an excellent job of modeling the activity of HVC neurons recorded in slice. All these results serve as the starting point for our study. We anticipate that the reviewer will have increased enthusiasm and reduced concerns regarding our paper based on this insight.

A related comment is that SK typically refers to a specific type of K⁺-sensitive Ca²⁺ channel.

Why is only apamin (SK channel inhibitor) tested and not iberotoxin (BK channel inhibitor). Do the authors know that BK channels are not present in HVCx cells? This seems odd as the presence of BK channels has apparently not been tested, or at least it is not described in the literature for this song nucleus.

Prior results demonstrate the presence of SK channels in HVCx. We were not aware of evidence that BK channels are present in HVCx neurons. We did not apply channel blocker drugs for channels that we did not know to exist.

As acknowledged in the paper, the higher similarities in spike properties of cells from sibling pairs as opposed to random pairs of birds could be due to either shared genetic factors or similarities in upbringing/learning. An obvious test that was not performed would have been a cross fostering experiment where sibling pairs are raised by different fathers.

Given that there are likely to be some genetic influences, as well as a predominant effect of song exposure, the full design is relatively complicated. To accurately assess the genetic contribution requires substantial cohort sizes and parallel experiments with non-related birds that are tutored the same songs. Given that the sibs were live tutored, then these experiments too should involve live tutored birds. This may seem simple at first glance but overall this is a substantial design beyond the scope of the present paper. Instead, we anticipate conducting these experiments over the next 2–3 years. Also, we demonstrated that IP are immature developmentally, so the proposed model will have the highly restricted IP space of a given adult somehow encoded but not expressed throughout development. Finally, Ross et al. 2017 and 2019 demonstrated that IPs are developmentally labile and sensitive to features of developmental song exposure.

With regards to the juvenile data and predictions from HH models, some further analysis of the current clamp data that is missing in the study could provide some relevant tests for the model inferences as to possible current types involved. For example, one would expect differences in the sag of the input resistance and/or in the AP half width or repolarization rate when comparing juveniles vs adults.

We did in fact report that there are differences in the sag current when comparing juveniles and adults (p. 15). Our observation is consistent with observations on juveniles and adults reported by Ross et al. 2017. As to AP half width or depolarization rates, these are features that characterize spike waveform shapes; we demonstrate that spike waveform shapes are different comparing juveniles with adults.

Given the acknowledged limitations in the dataset presented, the following is an overstatement that should be amended/toned down: "not via purely cell autonomous phenomena but in a manner sensitive to network and behavioral constraints."

We showed features of intrinsic properties are related to an animal's singing behavior and are rapidly modified by changes in the animal's singing behavior. Both our work and the work of the Hyson lab show these properties vary developmentally, and the recent Hyson lab paper demonstrates that song rearing affects HVCx intrinsic properties. Given this background we sincerely do not understand what limitations in the dataset require amending this very modest conclusion, so we do not know how to comply with the reviewer's request.

In the DAF experiment, apparently no characterization of the songs from the experimentally manipulated birds was performed. Is the assumption here that it was not necessary to characterize the birds' songs because no changes to the songs were expected, given the prior literature? However, given the apparent changes that occurred in the intrinsic properties of HVCx cells, and the correlation of these properties with song bioacoustics that is being proposed by the authors, it seems like it would be critical to directly demonstrate that these birds did not change their songs. To take this point further, if no changes actually occurred in these songs of the DAF exposed birds, it seems hard to reconcile such a finding with the main conclusion that the parameters of HVCx cells reflect song representation, as proposed in the title of the paper. Furthermore, if the songs of the exposed birds did not change, then the statement that the observed changes in the intrinsic properties of HVCx cells are due to learning does not seem well supported.

The reviewer may have overlooked the sentences where we reported changes in singing behavior after the onset of cDAF exposure. We now expand this description by providing some quantification of the effects. Please see response to major comment (13) of Reviewer 1.

Other specific comments:

Abstract:

...with non-synaptic mechanisms... Not clear from the manuscript where this conclusion arose from, it seems highly speculative at this point. The authors should be more explicit when they are discussing a cell's intrinsic properties vs. changes in properties that presumably arise from altered synaptic communication.

We have modified the abstract accordingly.

'Ballistic singing' is used several times but not defined in the text.

This is now defined.

The text states: ...all of which were clearly projecting towards Area X (Supplementary Fig. 1a), yet this figure does not depict axonal terminals in Area X, as the text seemed to imply.

We do not state or imply the figure depicts axonal terminals in Area X, only that we show axons in HVC projecting towards Area X following a well-established trajectory, and one that is distinct from the trajectory taken by RA-projecting HVC neurons (see Fortune and Margoliash J Comp Neurol 1995). In the figure (then Supp Fig. 1), the orientation of the filled cells was not adjusted to the orientation of song system diagram, which may lead to confusion. We have modified the figure to address this issue (now Fig. 1a, 1b).

'Plateau amplitude' seems to be defined only in Supplementary Figure 3, this should be made more clear in the main text and/or Figure 2 legend.

Please see response to major comment (7) of Reviewer 1. The figure was moved to new Supplementary Figure 3, where we also added examples of plateau amplitude.

It is unclear why some other important spike parameters were not analyzed, such as AP half-width, or Max repolarization and depolarization rates. Such parameters might have been informative when analyzing cases where changes in K⁺-related currents are suspected, such as in juveniles or to test some predictions from the modelling effort.

The HH modeling approach we used is sensitive to changes to AP half-width and max repolarization and depolarization rates since it matches the dynamics of the individual channel kinetics to the spike waveforms. It represents the combined effect or “emergent” features estimating all five conductances acting together dynamically. Therefore, when we report differences current magnitudes comparing juveniles and adults using the HH modeling this arises from differences in spike shapes (which we analyze). It is not clear to us that any inferences from extending the analyses to include changes in AP half-width and max repolarization and depolarization rates would yield inferences that would be stronger than what we have reported.

Did input resistance vary across birds in correlation with the observed differences in spike properties? This seems relevant since input resistance can have significant effects in shaping active neuronal cell properties.

Correct, thank you. Please see response to major comment (8) of Reviewer 1.

A related question is whether the observed variability as stated in the text (181 \pm 35 M Ω) refers to SD or SE?

We added a comment near the beginning of the paper that all measurements are reported as mean \pm SD unless otherwise stated.

With regards to the statement: "Model fits were judged based on the response to current pulses of 100 pA;", what is the typical magnitude current from synaptic input onto HVCx neurons? Is 100 pA reasonable or appropriate?

Current injections of < 50 pA rarely cause significant changes in HVCx firing patterns recorded in slice, whereas above 175 pA responses of HVCx can saturate. In good recordings of HVCx such as we report here, 100 pA yields consistent and reproducible results. Also, 100 pA is reasonable and appropriate in the sense that it is well within the norm for similar studies of HVC neurons in slice from numerous labs stretching over roughly 25 years.

With regards to the statement: "In our sample, model estimates of g_{Na} varied within the species by over 19.2x...", this seems like a very high level of variability, given that the range of variability in the 'first spike amplitudes' of actually recorded cells, as shown in Fig. 2c, is in the order of 35 mV or so. Is there any evidence that this would match variability in actual sodium currents recorded from real HVCx neurons?

We agree this is a high level of variability across animals. We are unaware of any direct measurements of sodium currents in HVCx, a study that our results now motivates. Potassium currents in homologous STG LP neurons in different animals vary by 2-4x, and it is not implausible that HVC currents would vary more (but not necessarily for the same ion current species) given the strong influences of learning mechanisms.

The authors do not provide a definition or explanation of how they determined the 'threshold' values shown in (Supplementary) Figure 3.

We now provide the missing definition of the spike threshold (the maximum of the second derivative of the voltage waveform) along with the definition of all other spike morphology/timing features in Supplementary Figure 2. The plot of spike threshold vs. first spike amplitude has been moved from supplementary material to Fig. 2c.

The recordings shown in (Supplementary) Figure 4 provide evidence for spontaneous firing of HVCx neurons, however this aspect was not discussed or examined in any detail. Is spontaneous firing a consistent feature of these cells? Is this an aspect that is predicted by the model?

In healthy HVC slices interneurons can exhibit considerable spontaneous activity, with lesser spontaneous activity observed for HVC_x and little (but non-zero) spontaneous activity for HVC_{RA}. We see this commonly. The spontaneous activity is likely to arise from network effects (dendritic input), for example see the results and discussion regarding blocking inhibition (Fig. 4). Our single-compartment model does not predict network effects.

There are several concerns with regards to the data presented in Supplementary Figure 7.. As mentioned above pairwise recordings with individual pharmacological inhibitors and a more rigorous statistical approach could be used to test individual currents that the authors have determined to play a role in HVCx neuronal physiology. What was the intersweep holding potential for the voltage clamp experiments in (B-D)? (D) should include a dashed line or some indicator for 0pA. The blue sodium current isolated in (D) does not have the profile of a typical sodium current as there is a very small, slow inward current. The authors should consider addressing this current's profile or replacing it with a better example.

We urge the reviewer to re-evaluate this comment having now recognized that the pharmacology of all the ion currents under investigation here was already rigorously established in the first author's prior studies.

Thank you, we have indicated 0 pA in Supplementary Fig. 8d (former Supp. Fig. 7). The holding potentials in Supp. Fig. 8 were stepped from -70 mV to -20 mV in Supp. Fig. 8b so that the intersweep holding potential was -70 mV between sweeps. Similarly, for Supp. Fig. 8c and 8d, the holding potential was -130 mV between sweeps.

Beyond that, please note that the small set of neurons shown in Supplementary Fig. 8 (old Fig. 7) were bathed in a complex cocktail of drugs, first to block fast synaptic transmission, and then with a set of ion channel blockers, and then later with additional sets of ion channel blockers, and for baseline and each condition were recorded in current clamp and voltage clamp modes. We observed a systematic pattern of results. At each stage of that experiment, the remaining currents yielded traces that were similar from different cells of the same bird and varied from bird to bird. This was a successful, independent test of a prediction that arose from the other aspects of the study. The reviewer is correct that only a small number of cells were recorded under the conditions of voltage clamp. However, these were the most difficult recordings reported here, requiring particularly lengthy recording times for each neuron, while holding the recordings under difficult conditions, and are at the extreme of the data set. We fully agree that the remaining current does not have the expected morphology of fast sodium currents. Perhaps

it represents another species of sodium current but in any case, we did not have the time to investigate this. We hope the reviewer can agree with us that the larger data set (and additional experimental manipulations) that is required to make a modeling study feasible is yet another good experiment that arises from the principal result we report here.

Alternatively, we can simply remove these supplemental data from the paper. They are not necessary for any principal claims of the study. They are, in our opinion, useful and yet supplemental information.

No description of the song bioacoustics analysis was presented in Methods.

Correct, thank you. This has now been added to the Methods, in Song recordings and in Delayed auditory feedback sections.

Minor:

Abstract:

... within hours if not quicker... : no support for 'if not quicker', this should be deleted from statement.

We modified the Abstract.

Typo: ...HVC spiking activity do not change in zebra finches...;

'Principal' component analysis, not 'principle'

The term IP is not defined in the text.

"N = 253 neurons arising from all adult birds with two or more HVCx neurons recorded"

All fixed.

Reviewers' Comments:

Reviewer #1:

Remarks to the Author:

The authors have addressed most of my comments in a satisfactory manner, and in cases in which this was not done, they have included principled reasons with which I broadly agree. The paper is ready for publication in my view.

Reviewer #2:

Remarks to the Author:

I find this to be an extremely interesting and novel study that will lead to many new directions in terms of thinking about how changes in intrinsic neuronal properties might be involved in organizing circuit dynamics and shaping (vocal) learning. The study is comprehensive in that it goes from describing the general description of the phenomenon (intrinsic neural properties within a specific song control area are more similar within an individual than across individuals), to modeling the underlying currents that might contribute to these intrinsic similarities to finally showing that this similarity is tied to song learning and that it can be disrupted by auditory feedback perturbation. Apart for a few minor issues in the text and the discussion, which I still find to be much less compelling than it could be, I do not have any major issues (I was already quite supportive the first time around) with this newly revised manuscript. This is an important finding.

Minor comments:

Figure 1c. The authors be more clear in the figure itself that each trace represents a different neuron within the same bird. One of the issues is that the figure legend is exceedingly long. The authors should consider shortening it.

Page 8, figure legend for figure 3. The authors make the statement that "Good fits resulted in good predictions in response to current injections not used in the fitting process, ...". Unless I missed it, I could not find any quantification for that statement which should be added if missing.

Page 16. In the sentence "... 6b, right panels), and this difference was highly statistically significant (see Methods, Additional statistical analyses, Fig 6e)", the authors state their findings are statistically significant. The statistical test does not appear to be indicated in either the legend or the results section.

Page 15, last paragraph. In the following statement "We observed clear effects of cDAF on singing behavior, preliminary observations that are consistent with more detailed observations made on other birds", it is not clear why the authors do not provide more detail regarding the effect of cDAF on song behavior. This seems like a critical part of the study and some degree of quantification needs to be provided given the claim the authors make.

cDAF experiments.

The authors should more clearly indicate whether songs manipulated with cDAF are from directed or undirected song.

Discussion:

The discussion is disappointing and does not convey the excitement of the study. The first paragraph for example, feels generic and vague. Furthermore, redundant statements such as "show developmental changes and variability during development" suggest that further editing could be used to tighten the language. Finally, many interesting points are raised by the other two reviewers. While I do not necessarily agree that these need to be added to the study, I do feel that the discussion would be a good place to address some of the issues that are raised.

Reviewer #3:

Remarks to the Author:

The authors have addressed many of the concerns raised in first submission, and the revised manuscript is substantially improved. This is a significant study that makes a substantial contribution to the field. Some remaining issues that have not been satisfactorily addressed, or that could be further improved in the paper, are detailed below. These do not detract from the main conclusions, but addressing them would help clarify some concerns and likely enhance the paper's potential impact.

Comment #1: The attempt to directly demonstrate/characterize the proposed individual currents in the recorded cells was very limited. More specifically, a more concerted effort is missing to record under voltage clamp to isolate and biophysically characterize specific currents, and to use paired recordings before and after the application of pharmacological blockers, including statistical analysis, to more clearly show that specific currents are sensitive to individually applied drugs that can affect spiking behavior. This would be an important complement to the modelling effort, but it is missing in the study. If the modeling was based only on data from the previous literature, this should be stated more explicitly, and limitations in data interpretation acknowledged accordingly.

Author's Response: Respectfully, we believe that this may represent a misunderstanding on the reviewer's part. As we stated in the paper, the ion currents that we modeled have been rigorously analyzed in prior work, see especially Daou et al. 2013 JNeurophys. There, among other results, authors used pharmacological approaches to demonstrate that each of the five ion current species are represented in HVCX neurons, and also developed the Hodgkin-Huxley modeling that is the basis for our analyses here. The single compartment HH model developed in Daou et al. 2013 did an excellent job of modeling the activity of HVC neurons recorded in slice. All these results serve as the starting point for our study. We anticipate that the reviewer will have increased enthusiasm and reduced concerns regarding our paper based on this insight.

Reply: The ability of the model to fit the experimental data is appreciated, as well as the fact that data from current clamp recordings in HVC from Daou et al., 2013 were used to inform the models used in the current study. The use of pharmacological blockers in that study allowed the authors to identify putative conductances in HVC neurons. However, these previous studies in finches did not measure biophysical parameters of these distinct ionic components of excitability of HVC in the voltage clamp configuration (i.e. activation, inactivation etc.). Therefore, the degree to which these parameters are similar or differ in finches compared to well-studied conductances that were the basis for the models utilized in Daou et al., 2013 remains unknown. Previous studies have shown that different combinations of conductances are capable of producing the same intrinsic excitable properties (Marder and Goaillard, Nature Reviews Neuroscience, 2006). Therefore, without further characterizing the biophysical properties of the distinct ionic currents that the authors claim are changing under a given manipulation, they must exercise caution in interpreting their results exclusively based on their models. The authors should further acknowledge these limitations of their study in determining how specific conductances change with their experimental manipulations.

Comment #2: A related comment is that SK typically refers to a specific type of K⁺-sensitive Ca²⁺ channel. Why is only apamin (SK channel inhibitor) tested and not iberotoxin (BK channel inhibitor).

Do the authors know that BK channels are not present in HVCx cells? This seems odd as the presence of BK channels has apparently not been tested, or at least it is not described in the literature for this song nucleus.

Author's Response: Prior results demonstrate the presence of SK channels in HVCx. We were not aware of evidence that BK channels are present in HVCx neurons. We did not apply channel blocker drugs for channels that we did not know to exist.

Reply: Just because no one has tested for BK it does not mean it's not there. Again, the authors should explicitly state limitations of modelling as detailed biophysical properties of zebra finch ionic currents do not appear to have been fully characterized.

Comment #3: With regards to the statement: "In our sample, model estimates of g_{Na} varied within the species by over 19.2x...", this seems like a very high level of variability, given that the range of variability in the 'first spike amplitudes' of actually recorded cells, as shown in Fig. 2c, is in the order of 35 mV or so. Is there any evidence that this would match variability in actual sodium currents recorded from real HVCx neurons?

Author's Response: We agree this is a high level of variability across animals. We are unaware of any direct measurements of sodium currents in HVCx, a study that our results now motivates. Potassium currents in homologous STG LP neurons in different animals vary by 2-4x, and it is not implausible that HVC currents would vary more (but not necessarily for the same ion current species) given the strong influences of learning mechanisms.

Reply: The authors could further address the wide variance of the NaV conductance and why they think there may be such a large variance considering the need for precisely timed AP firing in HVC neurons. How may these differences impact firing frequency in these HVC neurons?

Comment #4, with regards to voltage-clamp experiments:

Reply to Author's Response: The claim that waveforms from voltage clamp experiments were similar within individuals tested is appreciated. The data in Supplemental Fig. 8C shows whole cell currents that represent the repertoire of ion channels underlying the APs seen in current clamp experiments. These data could be expanded to include more individuals (authors only show 3) and more cells within individuals (Orange 261 has only 2 cells shown). These experiments do not require voltage gated ion channel blockers and should be easier than experiments isolating individual currents (As in Supplemental figure 8D). The experiments in Supplemental Figure 8D attempt to isolate only NaV currents. However they are just one of many ion channels the authors could have looked at. These currents do not seem to be well isolated with Orange 86 Nav currents seemingly non-existent. There are also very few animals (N=3) and cells (in some cases only 2) recorded from. The authors could attempt these experiments with more animals and more cells per animal. Alternatively, the recommendation would be to remove this data from the manuscript considering the difficulty of isolating individual current components from the Finch.

Reviewer #1 (Remarks to the Author):

The authors have addressed most of my comments in a satisfactory manner, and in cases in which this was not done, they have included principled reasons with which I broadly agree. The paper is ready for publication in my view.

Thank you.

Reviewer #2 (Remarks to the Author):

I find this to be an extremely interesting and novel study that will lead to many new directions in terms of thinking about how changes in intrinsic neuronal properties might be involved in organizing circuit dynamics and shaping (vocal) learning. The study is comprehensive in that it goes from describing the general description of the phenomenon (intrinsic neural properties within a specific song control area are more similar within an individual than across individuals), to modeling the underlying currents that might contribute to these intrinsic similarities to finally showing that this similarity is tied to song learning and that it can be disrupted by auditory feedback perturbation. Apart for a few minor issues in the text and the discussion, which I still find to be much less compelling than it could be, I do not have any major issues (I was already quite supportive the first time around) with this newly revised manuscript. This is an important finding.

Thank you. We have modified the discussion along lines which we believe make it more compelling, while responding to remaining issues (see below).

Minor comments:

Figure 1c. The authors be more clear in the figure itself that each trace represents a different neuron within the same bird. One of the issues is that the figure legend is exceedingly long. The authors short consider shortening it.

We clarified the description in the figure legend along the lines the reviewer suggests. None of the ways to shorten the figure legend that we attempted (focused on Fig. 1a) were satisfactory, and we abandoned the effort pending further advice.

Page 8, figure legend for figure 3. The authors make the statement that “Good fits resulted in good predictions in response to current injections not used in the fitting process, ...”. Unless I missed it, I could not find any quantification for that statement which should be added if missing.

We modified the language in the figure legend along the lines suggested. The reviewer may have overlooked the paragraph following the figure, which discusses quantification of the model fits at great length.

Page 16. In the sentence “... 6b, right panels), and this difference was highly statistically significant (see Methods, Additional statistical analyses, Fig 6e)”, the authors state their findings

are statistically significant. The statistical test does not appear to be indicated in either the legend or the results section.

The reviewer is correct, the statistical test is reported not in results or figure legend. It is reported in the Methods section (as we wrote, see above) which follows the end of the main text, in the subsection “Additional statistical analyses”, subpart Fig 6e (page 26).

Page 15, last paragraph. In the following statement “We observed clear effects of cDAF on singing behavior, preliminary observations that are consistent with more detailed observations made on other birds”, it is not clear why the authors do not provide more detail regarding the effect of cDAF on song behavior. This seems like a critical part of the study and some degree of quantification needs to be provided given the claim the authors make.

We provide quantitative analysis demonstrating that in the first few hours following onset of cDAF birds reduce the amount of singing and increase the number of introductory notes (Table 1). That is, their singing behavior is clearly and significantly affected. Also, we restrict our analysis to the first four hours of singing, supporting the observation of changes in IP of a bird that received only four hours of the abnormal feedback. We believe the reviewer is interested in further analysis of changes in HVC_X and changes in singing behavior which might yield additional insight into what changes in singing behavior are associated with what changes in HVC_X. If we had such results we would report them, but as of yet we do not.

The authors should more clearly indicate whether songs manipulated with cDAF are from directed or undirected song.

Done (p. 15).

Discussion:

The discussion is disappointing and does not convey the excitement of the study. The first paragraph for example, feels generic and vague. Furthermore, redundant statements such as “show developmental changes and variability during development” suggest that further editing could be used to tighten the language. Finally, many interesting points are raised by the other two reviewers. While I do not necessarily agree that these need to be added to the study, I do feel that the discussion would be a good place to address some of the issues that are raised.

We have taken up the charge of the reviewer, and revised parts of the discussion including the first paragraph. Since there was some skepticism and a number of issues that needed to be resolved with the reviewers, we had been hesitant to extend ourselves in the discussion. Hopefully this will now be well received.

Reviewer #3 (Remarks to the Author):

The authors have addressed many of the concerns raised in first submission, and the revised manuscript is substantially improved. This is a significant study that makes a substantial

contribution to the field. Some remaining issues that have not been satisfactorily addressed, or that could be further improved in the paper, are detailed below. These do not detract from the main conclusions, but addressing them would help clarify some concerns and likely enhance the paper's potential impact.

Comment #1: The attempt to directly demonstrate/characterize the proposed individual currents in the recorded cells was very limited. More specifically, a more concerted effort is missing to record under voltage clamp to isolate and biophysically characterize specific currents, and to use paired recordings before and after the application of pharmacological blockers, including statistical analysis, to more clearly show that specific currents are sensitive to individually applied drugs that can affect spiking behavior. This would be an important complement to the modelling effort, but it is missing in the study. If the modeling was based only on data from the previous literature, this should be stated more explicitly, and limitations in data interpretation acknowledged accordingly.

Author's Response: Respectfully, we believe that this may represent a misunderstanding on the reviewer's part. As we stated in the paper, the ion currents that we modeled have been rigorously analyzed in prior work, see especially Daou et al. 2013 JNeurophys. There, among other results, authors used pharmacological approaches to demonstrate that each of the five ion current species are represented in HVCX neurons, and also developed the Hodgkin-Huxley modeling that is the basis for our analyses here. The single compartment HH model developed in Daou et al. 2013 did an excellent job of modeling the activity of HVC neurons recorded in slice. All these results serve as the starting point for our study. We anticipate that the reviewer will have increased enthusiasm and reduced concerns regarding our paper based on this insight.

Reply: The ability of the model to fit the experimental data is appreciated, as well as the fact that data from current clamp recordings in HVC from Daou et al., 2013 were used to inform the models used in the current study. The use of pharmacological blockers in that study allowed the authors to identify putative conductances in HVC neurons. However, these previous studies in finches did not measure biophysical parameters of these distinct ionic components of excitability of HVC in the voltage clamp configuration (i.e. activation, inactivation etc.). Therefore, the degree to which these parameters are similar or differ in finches compared to well-studied conductances that were the basis for the models utilized in Daou et al., 2013 remains unknown. Previous studies have shown that different combinations of conductances are capable of producing the same intrinsic excitable properties (Marder and Goaillard, Nature Reviews Neuroscience, 2006). Therefore, without further characterizing the biophysical properties of the distinct ionic currents that the authors claim are changing under a given manipulation, they must exercise caution in interpreting their results exclusively based on their models. The authors should further acknowledge these limitations of their study in determining how specific conductances change with their experimental manipulations.

We have added a paragraph discussing the HH modeling and its limitations, addressing the reviewer's concerns.

Comment #2: A related comment is that SK typically refers to a specific type of K⁺-sensitive Ca²⁺ channel. Why is only apamin (SK channel inhibitor) tested and not iberotoxin (BK channel inhibitor). Do the authors know that BK channels are not present in HVCx cells? This seems odd

as the presence of BK channels has apparently not been tested, or at least it is not described in the literature for this song nucleus.

Author's Response: Prior results demonstrate the presence of SK channels in HVCx. We were not aware of evidence that BK channels are present in HVCx neurons. We did not apply channel blocker drugs for channels that we did not know to exist.

Reply: Just because no one has tested for BK it does not mean it's not there. Again, the authors should explicitly state limitations of modelling as detailed biophysical properties of zebra finch ionic currents do not appear to have been fully characterized.

Indeed SK refers to a specific type of K⁺-sensitive Ca²⁺ channel, which we explicitly chose because it had been previously identified as a principal current in HVC_x neurons. In principle BK channels as well as numerous others might be expressed in HVC_x neurons but have not yet been identified. We are not aware of any evidence of BK channels expressed in HVC neurons. Nevertheless, in response to the reviewer's comments we made preliminary experiments testing for BK currents in HVC_x slice recordings. We used the potent inhibitor iberiotoxin, finding no effects when bath applied to recordings of several HVC_x neurons and at two concentrations of iberiotoxin. Thus unless iberiotoxin does not inhibit BK channels in passerine birds (or at least zebra finches) the specific suggestion of the reviewer does not obtain. Inhibitors for the known channels expressed in HVC_x work in zebra finches as known in other systems.

The reviewer's comment also reflects a larger concern. We have attempted to respond to that concern in a new, second paragraph of the discussion including the comment "...as well as additional unknown ion channels that HVC_x might express...").

Comment #3: With regards to the statement: "In our sample, model estimates of g_{Na} varied within the species by over 19.2x...", this seems like a very high level of variability, given that the range of variability in the 'first spike amplitudes' of actually recorded cells, as shown in Fig. 2c, is in the order of 35 mV or so. Is there any evidence that this would match variability in actual sodium currents recorded from real HVC_x neurons?

Author's Response: We agree this is a high level of variability across animals. We are unaware of any direct measurements of sodium currents in HVC_x, a study that our results now motivates. Potassium currents in homologous STG LP neurons in different animals vary by 2-4x, and it is not implausible that HVC currents would vary more (but not necessarily for the same ion current species) given the strong influences of learning mechanisms.

Reply: The authors could further address the wide variance of the NaV conductance and why they think there may be such a large variance considering the need for precisely timed AP firing in HVC neurons. How may these differences impact firing frequency in these HVC neurons?

As the analyses in the paper show, we do not predict a simple relation between the variation in any given conductance and the learned song. Thus, pursuing an analysis such as the reviewer suggests could be valuable but is likely to yield only partial insight because it is done purely from a physiological perspective. It does not account for the influence of learning which we demonstrate is so influential on the properties of these HVC neurons. (In the language of physiology, it does not account for factors that covary among the conductances.) We are currently pursuing this analysis, which is more

complicated than the approach that the reviewer suggests. It is beyond the scope of the current paper.

In our experience modeling HVC_x, excitability (hence spike rate) is more likely to be driven by the SK current than the NaV current. To examine the reviewer's suggestion quantitatively, however, we calculated the estimates of NaV conductance vs. spike rate from the data reported in the paper. As the graphic demonstrates, there are hints of structure but there is no simple relation. Adding this to the paper would not be meaningful without our doing a great deal of additional work. This is the proper subject for a separate paper.

Comment #4, with regards to voltage-clamp experiments:

Reply to Author's Response: The claim that waveforms from voltage clamp experiments were similar within individuals tested is appreciated. The data in Supplemental Fig. 8C shows whole cell currents that represent the repertoire of ion channels underlying the APs seen in current clamp experiments. These data could be expanded to include more individuals (authors only)

show 3) and more cells within individuals (Orange 261 has only 2 cells shown). These experiments do not require voltage gated ion channel blockers and should be easier than experiments isolating individual currents (As in Supplemental figure 8D). The experiments in Supplemental Figure 8D attempt to isolate only NaV currents. However they are just one of many ion channels the authors could have looked at. These currents do not seem to be well isolated with Orange 86 Nav currents seemingly non-existent. There are also very few animals (N=3) and cells (in some cases only 2) recorded from. The authors could attempt these experiments with more animals and more cells per animal. Alternatively, the recommendation would be to remove this data from the manuscript considering the difficulty of isolating individual current components from the Finch.

We agree that the data set of these very difficult recordings as reported in Supp. Fig. 8 is limited. In our opinion they give important additional insight that is consistent with and extends our central hypothesis, and should be reported in supplementary material not the main text because of the limited sample size.

But, to no avail. If we conduct extensive additional recordings such as the reviewer suggests we will report them as the central result of a follow-up paper. We have withdrawn Supp Fig. 8 and the paragraph in the main text devoted to it, and modified references to the other supplementary figures accordingly.

Reviewers' Comments:

Reviewer #3:

Remarks to the Author:

The authors have satisfactorily addressed the reviewers' concerns, and the revised manuscript is substantially improved compared to previous versions. This is a significant contribution to the field and there are no remaining issues.